

# Impact of atmospheric turbulence on the accuracy of point source emission estimates using satellite imagery

Michał Gałkowski[1,2], Julia Marshall[3], Blanca Fuentes Andrade[4], and Christoph Gerbig[1]

[1]Department of Biogeochemical Signals, Max Planck Institute for Biogeochemistry, Jena, Germany
[2]Faculty of Physics and Applied Computer Science, AGH University of Kraków, Kraków, Poland
[3]Institut für Physik der Atmosphäre, Deutsches Zentrum fur Luft- und Raumfahrt, Oberpfaffenhofen, Germany
[4]Institute of Environmental Physics (IUP), Universität Bremen, Bremen, Germany

**Correspondence:** Michał Gałkowski (michal.galkowski@bgc-jena.mpg.de)

**Abstract.**

Observation-based monitoring of the status of greenhouse gas emissions goals set at the 2015 Paris Climate Summit is critical to provide timely, accurate and precise information on the status of the progress towards these goals. Observations also permit the identification of potential deviations from the adopted policies that could compromise the efforts to reduce future impact of pollutants on the climate.

Current remote sensing capabilities of atmospheric $CO_2$ have demonstrated the ability to estimate emission from the strongest sources of $CO_2$, based on imagery of single plumes combined with wind speed estimates. Realistically assessment of the accuracy and precision of the obtained emission estimates is critical, however. Here, we investigate the stochastic impact of daytime atmospheric turbulence on the estimations of $CO_2$ emissions from a lignite coal power plant in Bełchatów, Poland, using a high-resolution (400 m x 400 m x 85 levels) atmospheric model set up in a realistic configuration. We show how the persistent structures in the emitted plumes cause significant uncertainties in retrieved fluxes when applying a commonly-used cross-sectional mass-flux method. on the order of 10 % of the total source strength. These form a significant contribution to the overall uncertainty which remains unavoidable in the presence of atmospheric turbulence.

Furthermore, the use of novel temporally-tagged tracers allowed for the decomposition of the plume variability into its constituent parts and explain why spatial scales of variability in plume intensity are far larger than the size of turbulent eddies – a finding that challenges previous assumptions.

## 1 Introduction

The importance of greenhouse gases (GHGs) for the Earth's climate, in particular $CO_2$, has been established for decades now. Their emissions to the atmosphere remain high and, more importantly above the optimal pathway that would assure limited climate change, represented by 1.5 °C mean atmospheric temperature increase by the end of 21[th] century against a 1850–1900 baseline (IPCC, 2023). Already, the warming results in multiple adverse effects across numerous domains of the Earth's System, causing not only damage to the environment humans live in, but directly affecting communities through



the strengthening of extreme events such as heatwaves, heavy precipitation, droughts and tropical cyclones. Furthermore, if unmitigated, the consequences of the increased $CO_2$ load will be felt for centuries to come.

Over the past decades, a range of international policies have been adopted aiming at minimising the adverse effects of climate change, with the 2015 Paris Agreement (UN, 2015) being the most recent effort coordinated within the United Nations frameworks. Mitigation plans within the Agreement are tightly connected with the annual reporting of the emission rates through National Inventory Reports (NIR), prepared in order to provide accurate, precise and timely information to the international community. Common methodologies used within NIRs are based on bottom-up statistical methods that in many
cases rely on indirect (proxy) datasets characterised by varying degrees of accuracy. Procedures for correcting inconsistencies have been developed, however the availability of accurate data is simply insufficient to provide estimates with low enough uncertainty, especially in underdeveloped countries. This affects our ability to formulate informed mitigation strategies. Therefore provision of transparent and accurate NIRs has a critical impact on building and maintaining society-wide trust, which is especially important as emission reduction and climate mitigation policies can incur monetary and societal costs. Independent,
science-based observations of GHG emissions provide a promising avenue that strengthens the confidence of all interested parties.

Moreover, monitoring of atmospheric greenhouse gases through both in situ and satellite-based measurements, coupled with modern top-down frameworks, can also help in reducing the uncertainty of the anthropogenic fluxes on the global scales, estimated at $9.6 \pm 0.5 \, \mathrm{GtC \, yr^{-1}}$ in the last decade (Friedlingstein et al., 2023). It also has been identified as a source of
complementary information to the national inventory data, which are based on annual statistics of human activities (Maksyutov et al., 2019). Increasingly accurate regional budgets are one of the most promising avenues towards reducing the remaining global budget imbalance, with the additional benefit of reducing the detection delay if emission patterns change. Peters et al. (2017) have shown that a decrease of 1% of the global emissions can be detected using the currently available tools only after a decade of observations. Accurate regional carbon budgets could significantly reduce this time, and, consequently, allow for a
faster reaction of policymakers. Furthermore, top down frameworks have also been recognized

Two issues affect our ability to provide accurate regional budgets of GHGs. First, ground-based networks do not have sufficient density to feed the models that would help accurately estimate the anthropogenic fluxes at sufficient accuracy where the strongest emissions occur (US, EU, China, India). Second, in developing countries, where the emissions are smaller but are characterised by larger uncertainties, the ground-based observations are either sparse or altogether missing. Airborne and
spaceborne platforms have an important role in filling the observation gap, as they can provide accurate information on high spatial resolution in the regions where only limited (or no) ground-based observations are available. Because of the limitations in terms of range, high unit costs and sparse temporal coverage, the relevance of airborne observations for direct emission estimation on the global or regional scales has historically been largely limited to constraining natural fluxes (Gerbig et al., 2003) or validation of global models (Gałkowski et al., 2021), but recent years have also brought robust studies covering
larger regions, especially for oil and gas related emissions (e.g. Sherwin et al., 2024). Nevertheless, airborne observations of in situ mole fractions have been successfully used to provide important insights into the sub-regional and local sources of anthropogenic GHG emissions, employing either pure data-focused analysis (Lowry et al., 2001; Turnbull et al., 2011), mass-



balance estimations (Klausner et al., 2020; Fiehn et al., 2020) or formal inversions of varying complexity (Krings et al., 2018; Lopez-Coto et al., 2020; Kostinek et al., 2021).

Rapid developments in remote sensing instrumentation have opened the avenue for direct estimations of GHG emissions. While some instrumentation has successfully been applied on airborne platforms (Krings et al., 2013; Krautwurst et al., 2021; Wolff et al., 2021), observations from orbiting platforms have a clear advantage thanks to their global coverage and lower per-observation cost. In fact, the newest generation of spaceborne sensors has already demonstrated the ability to estimate emissions of pollutants from larger emitting regions and also from single sources - if sufficiently strong. For example, OCO-2/3 were used

to estimate $CO_2$ emissions from selected large cities and power plants (Nassar et al., 2017; Reuter et al., 2019; Fuentes Andrade et al., 2024), and GHGSat-D has shown promise in detecting localised $CH_4$ plumes with rapid emission estimation (Jervis et al., 2021). These successful deployments further motivate the development and usage of an operational chain of dedicated satellite missions. Early works describing such a system, initially envisaged as CarbonSat (Bovensmann et al., 2010), have been subsequently expanded and resulted in the design and approval of the CO2M (Copernicus Anthropogenic $CO_2$ Monitoring

Mission), a constellation of satellites that are to be launched within the current decade (Sierk et al., 2021).

       A variety of methods have been applied to estimate GHG emissions using remote sensing observations, with a good general description of their assumptions and respective strengths and weaknesses available in Varon et al. (2018, also references therein). From the four estimation methods listed, Gaussian Plume Inversion (GPI, Krings et al., 2011) and Nassar et al. (2017)), Integrated Mass Enhancement (IME, Frankenberg et al., 2016) and the Cross-Sectional Flux method (CSF, Krings et al., 2011)

have been most prominently used in practical applications the past. More recently reported developments include improvement of plume detection algorithms, also by using auxiliary $NO_2$ measurements (Kuhlmann et al., 2019, 2021), analysis of estimation statistics from repeated scenes by single spaceborne instrument (Nassar et al., 2022; Fuentes Andrade et al., 2024), and using detailed bottom-up information for comparisons (Fuentes Andrade et al., 2024). However, across those multiple studies, even when employing the available methods in the most favourable conditions, i.e. cloud-free atmosphere with negligible gradients

in the background fields, for point sources (like power plants) the reported uncertainty of emissions estimated from a single overpass remains between 10 % to 20 % most of the time, with significant variability stemming from the estimation method employed, wind speed, stability conditions, time of day etc.

       A significant contribution to the reported uncertainty stems from spatial variability not accounted for explicitly in any of the methods if applied to spaceborne or airborne data, due to stochastic turbulence present in the daytime atmosphere, when

virtually all relevant observations have been collected so far. The first analysis of this spatial variability in atmospheric $CO_2$ was applied to the vertical distribution of $CO_2$ in Gerbig et al. (2003) and augmented by Lin et al. (2004) to assess representation errors associated with the spatial grid resolution of transport models typically used in inverse modelling. A similar analysis has been used to estimate the uncertainty of emission estimates in a recent study by Fuentes Andrade et al. (2024), demonstrating the variability of $CO_2$ emission estimates using the CSF method (Fig. 1).

Here, a corresponding method is deployed to assess scales of variability on somewhat higher spatial resolution, as apparent in partial columns in simulated plumes of $CO_2$, to assess the impact on the uncertainty of point source emissions and provide insight into detailed mechanisms influencing emission estimates from the CSF method. In order to shed some light on



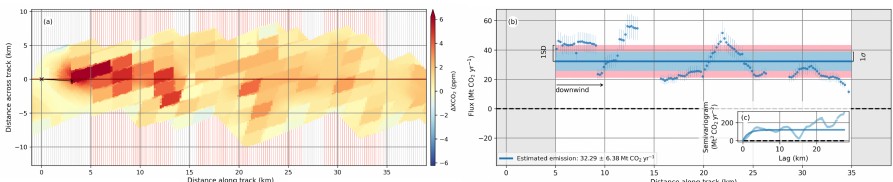

**Figure 1.** $CO_2$ plume from Bełchatów Power Plant observed by OCO-3 on 10 April 2020. Left: wind-corrected OCO-3 observations. Right: Apparent emissions estimated using CSF method. Adapted from Fuentes Andrade et al. (2024).

these phenomena, we are using high-resolution WRF-GHG simulation over a previously studied point source, augmenting the modelling system with temporally-tagged tracers.

The paper is structured as follows: Section 2 is dedicated to the description of the experimental setup, study area and model configuration. A detailed description of the temporally tagged tracer concept and application is provided as well. Section 3 presents the results and Sect. 4 their detailed discussion. The conclusions and outlook are presented in Sect. 5.

## 2    Methods

### 2.1    Study area

The Bełchatów Power Plant (BPP) is one of the largest anthropogenic point emitters of $CO_2$ on the entire planet, using lignite coal for power production. The nominal capacity of the plant was 5102 MW of electrical power in 2021, approximately 13 % of the total capacity of the Republic of Poland[1]. Under legislation at both national and EU levels, accurate information on GHG emissions and operational status is publicly available, making it an excellent target for the development and testing of new instruments and methods. In fact, BPP has already been used in several studies focusing on developing emission estimation
methods (Nassar et al., 2022; Fuentes Andrade et al., 2024) or modelling approaches (Brunner et al., 2023).

The power station is located at 51.267°N, 19.325°E, in the vicinity of Bełchatów in Central Poland. Emissions of $CO_2$ and other compounds are reported through the European Pollutant Release and Transfer Register (E-PRTR). Reported emissions of $CO_2$ from 2018 to 2022 varied between $30.1\,\mathrm{MtCO_2\,yr^{-1}}$ and $38.4\,\mathrm{MtCO_2\,yr^{-1}}$, with a minimum in 2020 (EEA, 2023). The topography of the surrounding area is mostly flat and characterised by minor orographic variability, with the notable exceptions
of the deep (up to approximately 200 m) open-pit lignite coal mine located directly to the south of the power plant, neighboured by the coal heap containing the residue of the mining operation (up to 175 m high) to the southeast (Fig 2). The area of the mine pit was approximately $12\,\mathrm{km^2}$ in 2020.

---

[1]https://pgegiek.pl/Nasze-oddzialy/Elektrownia-Belchatow, last access 30.08.2024



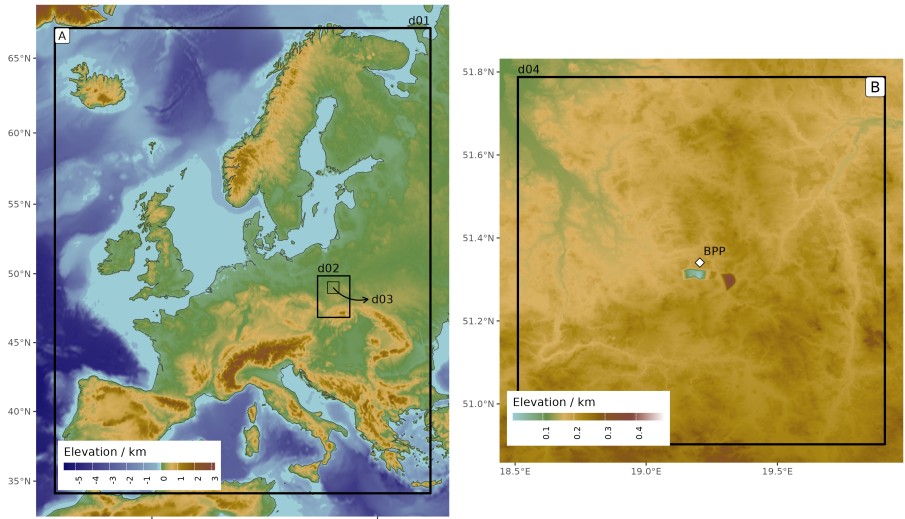

**Figure 2.** WRF domains for the simulations superimposed on a topography map. A – extent of parent and nested domains. B - high-resolution domain. BPP – Bełchatów Power Plant, location marked with white rhombus.

## 2.2 WRF-GHG

The numerical experiment presented here was performed using the Weather Research and Forecast Eulerian model WRF (Ska-
marock et al., 2008) with the Advanced Research WRF (ARW) core enabled. WRF was developed within a large collaborative
project led by the National Center for Atmospheric Research (NCAR), and has been augmented over the years by improvements
from a number of community users. The model integrates the non-hydrostatic, fully compressible flux-form Euler equations on
a terrain-following mass-based vertical coordinate, and has been successfully applied for meteorological and tracer-transport
studies at scales ranging from global to local scales, thanks to the ability to dynamically downscale the computations through
a nesting algorithm.

For our experiment, we deployed WRF v3.9.1.1. with the addition of the GHG module (Beck et al., 2011), implemented
within the WRF-Chem software package (Grell et al., 2005; Ahmadov et al., 2009). Hereafter we refer to this framework as
WRF-GHG. The module allows for the emission, transport and mixing of inert $CO_2$ tracers, as well as online calculation
of photosynthetic and respiration fluxes, although that feature was not used in the current study. We applied the system in a
limited area mode, using meteorological boundary conditions from the ECMWF Integrated Forecasting System HRES run,
downloaded at $0.125° \times 0.125°$ horizontal and L137 vertical resolution ECMWF (2022).

The model was run in a one-way nested configuration with three domains of gradually increasing spatial and temporal
resolution (Fig. 2). The parent domain spanned continental Europe with a 10 km horizontal grid. The intermediate nested
domain covered parts of southern and central Poland at 2 km horizontal resolution and the final nested domain was run at 400
m horizontal resolution and spanned a rectangular area of 100 km x 100 km, centred around BPP. To ensure model stability,



the domains were run with time steps of 50 s, 10 s and 2 s, respectively. We used the classical mass-based terrain-following $\eta$ coordinate definition, with the model top set at a constant p = 50 hPa, corresponding to approximately 20 km amsl. As the vertical transport of tracers was one of the key phenomena investigated in our experiment, we have also used a high-resolution vertical level structure in the lower atmosphere, with 85 full-model levels between the surface and the model top. The lowest

layer thickness was set to 25 m, with 38 levels below 3 km altitude.

We have used WRF parameterizations suitable for the spatio-temporal scales involved, including the Thompson microphysics scheme, RRTMG schemes for longwave and shortwave radiation, the revised MM5 scheme for surface layer physics and the Noah Land Surface Model. Grell 3D cumulus parameterization was enabled in the parent domain only. Full model settings (including all relevant citations) are provided in Table S1. As the nested domains are run at horizontal resolutions in

the so-called "grey zone" (i.e. grids with horizontal spacing between 0.2 km and 6 km; see Honnert et al., 2020), we have applied the Shin-Hong PBL (Planetary Boundary Layer) scheme (Shin and Hong, 2015) for all simulated domains. This parameterization introduces scale dependency for vertical transport in the convective PBL and follows the YSU scheme in the free atmosphere. We have used the default MODIS land use category maps (at 30") and elevation maps (at 1' resolution for the parent domain and 30" for nested domains). Grid nudging was applied in the parent domain to maintain wind, temperature

and moisture fields consistent with the driving meteorological data at continental scale. We did not apply any nudging to the intermediate and high-resolution domains to allow the WRF internal parameterizations to drive the tracer transport at smaller scales. The strength of the nudging coefficient for water vapour was reduced to $4.5 \times 10^{-5} \, \mathrm{m \, s^{-1}}$ following Spero et al. (2018). Instantaneous model output was saved every five minutes. Comparisons to observational data were performed using the output with the minimal time difference.

Prior to data analysis, simulated $CO_2$ fields were interpolated from WRF's Lambert Conformal Conic projection to a Cartesian coordinate system centred at BPP and oriented towards the direction of the effective wind ($u_{\mathrm{eff}}$), calculated every minute as an average of local wind speeds sampled between 200 m–600 m agl over a square area (20 km x 20 km) surrounding BPP. The height range was selected as the applied emissions are distributed mostly in this range (see the following section).

The vertical structure of the original WRF grid was preserved exactly, while the horizontal resolution was increased twofold

in order to better preserve the spatial features of the modelled plume (200 m × 200 m), using bilinear interpolation. The output grid formed a perpendicular area ranging from -5 km to +40 km in the X (along-wind) direction and -25 km to 25 km in the Y direction, in order to capture the full width of the plume throughout the period relevant for the analysis.

### 2.3 Emissions & Tagged tracers

For simplicity, we treated the plant as a single-point source, which is a good assumption considering our horizontal grid size

of 400 m. We applied emissions of $CO_2$ at the constant rate equal to the average annual emissions officially reported by BPP for the year 2018, i.e. 38.4 $\mathrm{MtCO_2 \, yr^{-1}}$ (EEA, 2023). Instead of a dedicated plume rise mechanism, we applied an invariable vertical profile in emissions, with tracer mass distributed along a Gaussian curve centred at $H_{\mathrm{eff}} = 4/3 \, H$, with a standard deviation of $\sigma_H = 1/3 \, H$, where H is the emitting stack height of 300 m. Examples of emission profiles and resulting model mole fractions are given in Fig. S1 in the supplement.



The emissions from the stack are and subsequently advected in the model throughout the full simulation period. We used this tracer primarily to monitor the spatial extent over which the tracers represent the whole plume, and to calculate the dependence of emission estimate statistics over variable distances. We also used 60 additional tracers tagged by the time of release (temporally-tagged tracers) in order to study the effects of atmospheric turbulence on source estimation inference. Each tracer corresponds to a short (three-minute) segment of the emitted plume. Resulting $CO_2$ signals are similar, in a sense, to

"particles" or "air parcels" considered in Lagrangian models. In the following, we refer to these emitted plumes as "puffs". Distribution of $CO_2$ mole fractions for selected puffs is presented in Fig. S2 in the supplement.

## 2.4   Simulated case

For our study, we ran the simulation for a period between 9 April 2020, 18:00 UTC and 10 April 2020, 21:00 UTC. 60 puffs were then emitted on 10 April 2020, between 09:00 and 12:00 UTC (11:00–14:00 LT) in successive three-minute periods, so

that the sum of these emissions corresponds to the full signal emitted from the stack over that time window.

    The numerical analysis of the output was performed when the final tracer was emitted completely, i.e. at 12:00 UTC. By that point, the oldest tagged tracers had already been advected through the modelling domain for three hours. We stored the 1-minute output for the high-resolution domain from 09:00 UTC until 21:00 UTC for maximum temporal coverage of the analysed day.

## 2.5   Column-averaged mole fractions

The column-averaged dry-air mole fraction, commonly retrieved from remote sensing measurements, is a scalar quantity proportional to the total mass of the tracer within the whole atmospheric column. For every output time, WRF-GHG provides 3D fields of dry-air mole fraction enhancements (designated as $\Delta C$ for the full plume tracer), from which we calculate column-averaged dry-air mole fraction using the following formula:

$$\Delta \mathrm{X}C_{ij} = \sum_k \Delta C_{ijk}\, \omega_{ijk}. \tag{1}$$

Here, $\Delta \mathrm{X}C_{ij}$ is the enhancement of the column-average dry-air mole fraction of $CO_2$ at coordinate $(x_i, y_j)$ of the Eulerian grid. $\Delta C_{ijk}$ is the dry-air mole fraction of $CO_2$ at model grid coordinates $(x_i, y_j, z_k)$ given in $\mathrm{mol\,mol^{-1}}$, and $\omega_{ijk}$ are the weights applied to each value, calculated as:

$$\omega_{ijk} = \frac{[n_d]_{ijk}}{\sum_k [n_d]_{ijk}} = \frac{1}{N_d}[n_d]_{ijk}. \tag{2}$$

Here, $[n_d]_{ijk}$ is the number of moles of dry air at $x_i, y_j, z_k$, and $N_d$ is the total number of moles of dry air throughout the air column. The formulas above are independent of axis orientation, but the values discussed are in the wind-rotated coordinate system, with X axis oriented along the wind direction.

    It should also be noted that when comparing model output against spaceborne measurements, such as discussed by Varon et al. (2018), $\Delta \mathrm{X}C_{ij}$ should be calculated for the full atmospheric column spanning from Earth's surface all the way to orbit,



where the density of air can be neglected. However, as a) in the present study we're not comparing against actual measurements and b) we limit the analysis of the apparent emission of a tracer which never leaves our computational domain through its top boundary, it is not necessary to extend our vertical profile beyond the model top at 50 hPa.

## 2.6   Cross-sectional flux method

By assuming that the mass of the tracer is conserved (true in the case of long-lived greenhouse gases advected over short

distances), emission rates at the source can be inferred from the mass passing through a downstream cross-sectional area. This mass-balance can be described mathematically as:

$$\Phi(x) = u_{\text{eff}} \int\limits_{-\infty}^{\infty} \Delta\Omega(x,y) \, dy \qquad (3)$$

Here, $x$ and $y$ denote coordinates (in m) in the rotated Cartesian grid, with the X axis oriented along the wind direction. $\Phi(x)$ denotes the estimated emission (further referred to as the "apparent emission") at cross-sections computed at $x$ (in $\text{kg s}^{-1}$).

$\Delta\Omega(x,y)$ is the column-averaged enhancement of $CO_2$ (in $\text{kg m}^{-2}$) integrated along the Y axis and $u_{\text{eff}}$ is the effetive wind speed in the direction along the X axis (given in $\text{m s}^{-1}$). Note that, because we aim to reproduce processing as performed in studies using actual satellite imagery, we assume that the wind is constant throughout the area of interest, and thus use the formula with $u_{\text{eff}}$ given by Varon et al. despite having access to complete modelled wind fields. This aligns with past studies, where the wind is often taken from coarser-resolution reanalysis fields, most of which do not represent variabilities on scales

below 30 km.

It can be shown that for the WRF Eulerian grid, $\Delta\Omega(x,y)$ can be discretized as:

$$\Delta\Omega_{ij} = \frac{\mu N_d}{A} \Delta X C_{ij} = \frac{\mu}{A} \sum_k \Delta C_{ijk} [n_d]_{ijk} \qquad (4)$$

where $\mu$ is the molar mass of $CO_2$ ($44.01 \text{ g mol}^{-1}$), $A$ is the horizontal model cell area in $\text{m}^2$ and other symbols are as before.

Applying the above into Eq. 3 in its discrete form yields the apparent emission at a given distance $x$ as:

$$\Phi(x_i) = \frac{\mu \, u_{\text{eff}}}{\Delta x} \sum_{j,k} \Delta C_{ijk} [n_d]_{ijk} \qquad (5)$$

where $\Delta x$ is the dimension of the model cell along the X axis (in meters), and other symbols are as before. By calculating the sum over a wide range of cross-wind distances (Y axis, index $j$), we made certain that the full plume extent is represented in our interpolated fields. Similarly, we also ensured that the plume is fully represented in the vertical direction (index $k$) over the analysis area.

Similar to Varon et al., we are using a single value for the effective wind speed $u_{\text{eff}}$, with a notable difference that we calculate it from wind fields sampled at altitudes close to the emission altitude, for which we use the assumed emission profile to select the appropriate vertical level of the wind sampling. This approach is a hybrid of those used in the recent studies of Kuhlmann et al. (2021) and Nassar et al. (2022). In the first study, the mean wind speed was calculated from the model output



winds, weighted by relative emission strength. The emission profile, however, was based on statistically averaged profiles and did not take into account actual stack heights. In the second, Nassar et al. took into account the stack height and then used the Gaussian-plume assumptions to estimate the plume centerline altitude ($H_{eff}$), and then used winds from reanalysis products extracted at the same height over the emission point. Here, we are calculating $u_{eff}$ as an average of wind speed values at altitudes between $H_{eff} \pm 2\sigma_H$ (200 m–600 m). To avoid very local wind variability affecting the $u_{eff}$, we also spatially averaged the wind speeds over a square area of $\pm\,20\,\mathrm{km}$ around the emission point, which mimics the effect of using a coarse-resolution reanalysis wind dataset like ERA5 (as in Nassar et al.).

When discussing correlations, we also make use of the normalized apparent emission anomaly $\lambda_\Phi(x)$ defined as:

$$\lambda_\Phi(x) = \frac{\Phi(x)}{\overline{\Phi}} - 1 \tag{6}$$

where $\overline{\Phi}$ is the average of $\Phi(x)$ calculated over selected $x$ range.

### 2.7 Effective number of observations and uncertainty of emission

To estimate the mean uncertainty of the apparent flux, we calculate the mean value of $\Phi$ from individual cross-sectional flux estimates. Due to the existence of autocorrelation in the $CO_2$ enhancement on short spatiotemporal scales, the uncertainty of the mean apparent emission $u(\overline{\Phi})$ is therefore also spatially correlated and the standard formula for type B uncertainty ($u_a$ as in eq. (3) in Zięba, 2010), needs to be modified by a factor dependent which is on the effective number of observations $n_{eff}$:

$$u_a(\overline{\Phi}) = u(\overline{\Phi})\sqrt{\frac{n-1}{n_{eff}-1}} \tag{7}$$

where the index $a$ denotes that the uncertainty is calculated for an autocorrelated sample of n observations. The number of effective observations can be calculated using the autocorrelation function (ACF) following the formula from Zięba (2010):

$$n_{eff} = \frac{n}{1 + 2\sum_{\kappa=1}^{n-1}\frac{n-\kappa}{n}\mathrm{ACF}_\kappa} \tag{8}$$

where $\kappa$ is the lag index of the discrete ACF function.

A similar approach to calculating uncertainty for correlated $CO_2$ data was applied by Gerbig et al. (2003) for airborne vertical profile data, and recently by Fuentes Andrade et al. (2024) to estimate the dispersion error component in remote-sensing-based flux estimates.

### 2.8 Decomposition of variability into contributions

To further study the plume dispersion dynamics, we are calculating and using two additional auxiliary variables: the location of plume centroids along X axis, and the wind speed at the time and location of emission. Their respective definitions and purposes are given below.



### 2.8.1 Plume centroids

Plume centroids can be considered an analogue of particle locations as used in atmospheric Lagrangian transport models. We define a plume centroid as the first moment of the distribution of the tracer's mole fraction, thus approximating each tracer's centre of mass. The location of the p-th tracer centroid along the X axis is calculated as:

$$\overline{x_p} = \frac{\sum_{ijk} \Delta c^p_{ijk} \, x_i}{\sum_{ijk} \Delta c^p_{ijk}} \tag{9}$$

Here, $\Delta c^p_{ijk}$ denotes the $CO_2$ enhancement of a single tagged tracer at coordinates $x_i, y_j, z_k$, defined as before. The location of the tracer centroids along the Y axis, $\overline{y_p}$ is calculated analogously.

In subsequent correlation analysis we have approximated the density function of puff centroids along the X axis by using a cubic spline interpolation of the binned number of centroids (2 km), as the sample size would be insufficient otherwise. We then used these to calculate the average and normalized anomaly of centroid numbers, denoted as $\overline{x_c}$ and $\lambda_c$, respectively. The latter was calculated analogously to Eq. (6).

### 2.8.2 Wind speed at emission point and time

The second auxiliary variable is calculated to study how the local, highly variable wind speed at the emission location and time affects the estimates of $\Phi$. For this purpose, we calculate wind speed at emission location: $WS_{emi}(x_i)$ (WS is used to avoid confusion with $u_{eff}$). $WS_{emi}(x_i)$ is meant to link the apparent emissions with the local wind speed at the time and location of emission. Because the signal in the plume is efficiently mixed, the local wind conditions at the emission point can only be monitored using puffs, thus each puff is allocated a mean wind speed at the stack during its time of emission ($\overline{WS_{emi}}^p$), and as the puffs are advected along the plume, the $WS_{emi}(x_i)$ can be calculated for any given plume element by averaging across its constituent puffs, using their respective mole fractions as weights.

More precisely, for each puff, $\overline{WS_{emi}}^p$ is calculated as the average of the $u$ wind component (i.e. parallel to the X axis) at the emission point over the emission time of each tracer (3 mins), over the vertical extent surrounding the stack height:

$$\overline{WS_{emi}}^p = \sum_k \frac{\overline{u_p}(z_k)}{n_z}. \tag{10}$$

Here, $\overline{u_p}(z_k)$ denotes three-minute averaged values of the parallel wind component (in the x-direction) at the stack horizontal coordinates, extracted at altitude $z$ for tracer $p$. We calculate the average over the vertical range over which the maximum emissions occur, i.e. between 200 m and 600 m above ground, and $n_z$ denotes the number of model levels whose centres fall within that range.

In order to link individual puff-values with the apparent emission downwind from the emission point, we use a two-step algorithm. First, for each spatial point in our model, we calculate the mean value of $\overline{WS_{emi}}$ for that point, weighted by the column-averaged dry-air mole fractions of each:

$$\overline{WS_{emi}}(x_i, y_j) = \sum_p \overline{WS_{emi}}^p \, \omega^p_{ij}, \tag{11}$$





where:

$$\omega_{ij}^p = \frac{\Delta X c_{ij}^p}{\sum_p \Delta X c_{ij}^p}$$

In the second and final step, we calculate the cross-section average $\mathrm{WS_{emi}}(x_i)$ (overbar omitted), by weighting by the $\mathrm{XCO_2}$ column-averaged mole fractions:

$$\mathrm{WS_{emi}}(x_i) = \sum_j \overline{\mathrm{WS_{emi}}}(x_i, y_j)\, \psi_{ij}, \tag{12}$$

with the weights $\psi_{ij}$ calculated as:

$$\psi_{ij} = \frac{\Delta X C_{ij}}{\sum_j \Delta X C_{ij}}$$

In the subsequent analysis, we also correlate $\mathrm{WS_{emi}}(x_i)$ with $\Phi(x)$ in order to assess the imprint of the initial wind speed on the apparent emissions estimated downwind of the source. To avoid potential numerical noise caused by calculating ratios for
low values, we limited the analysis to those model grid points for which the total simulated $CO_2$ enhancements were higher than $0.01\ \mathrm{\mu mol\, mol^{-1}}$.

## 3 Results

### 3.1 Simulated plume structure

Our simulations demonstrate that the meteorological conditions on the day of interest were typical of a mid-latitude, clear-sky
spring day, with a nocturnal stable atmosphere evolving into a turbulent PBL over the course of the morning (05:00–11:00 UTC) and by 11:00 UTC the turbulence in the lower atmosphere has already been established. In our analysis, we focused on the state of the atmosphere at 12:00 UTC (14:00 CEST), 1 hour and 14 minutes after the local solar noon (10:44 UTC on 10 April 2020), when the PBL was already well developed and the $CO_2$ plume emitted from BPP had been advected and mixed for several hours. This is consistent with typical observation times of passive remote sensing instruments operated on platforms
orbiting on sun-synchronous orbits, as overpass times near local solar noon provide a high signal-to-noise ratio.

Figure 3 presents a set of cross-sections of the simulated $CO_2$ plume at 12:00 UTC. As can be seen, the model simulates a turbulent plume to a distance of 40 km from the emission source, with significant dispersion in both the horizontal $(x, y)$ as well as vertical $(z)$ directions. High mole fractions close to the emission source, reaching around $1000\ \mathrm{\mu mol\, mol^{-1}}$, are quickly dispersed by advection and turbulence, dropping to below $60\ \mathrm{\mu mol\, mol^{-1}}$ already 5 km downwind from the source,
and are further reduced as dispersion spreads the mass of tracer perpendicular to the main wind direction. Of note is that the model predicts very efficient vertical mixing of the emitted tracer from the surface to the top of the PBL (located at 1.6 km at 12:00 UTC). When averaging the plume along the X axis across the whole analysis area, the tracer mass is distributed almost uniformly up to the PBL top, distributed primarily around $y = 0$ across the plume, with a skew towards positive $y$ values (Figure 3, lower right).





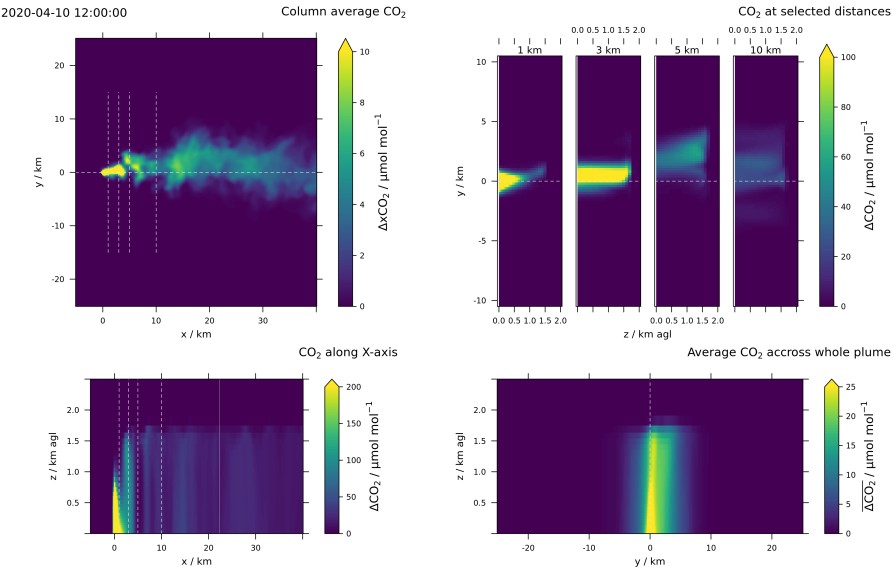

**Figure 3.** Simulated plume structure on 12:00 UTC, 10 April 2020. Upper-left: column-average $CO_2$ emitted from BPP. Cross-sections presented on the right are marked with dashed lines. Upper-right: $CO_2$ enhancements simulated at cross-sections located 1, 3, 5 and 10 km downwind from the emission source. Lower-left: cross-section of mole fractions sampled along the X axis (along wind). Lower-right: average $CO_2$ mole fraction enhancement in the Y-Z plane calculated across all distances (-5 to 40 km).

We compared the emitted total-$CO_2$ tracer with the sum of the 60 puffs, to make certain that no notable differences in the overall plume structure are caused by the numerical effects of the WRF advection schemes applied. These occur due to different gradients present in the tagged and total tracer fields. At 12:00 UTC, which marks the end of the period of puff emissions, the plume is fully represented at distances from the emission point at 0 km down to approximately 22 km. Local discrepancies between the sum of tagged tracers and the classical full tracer are caused by the advection scheme. Point-wise differences of

mole fractions of the two plume realisations can reach as high as 1000 $\mu mol\,mol^{-1}$ in the immediate vicinity of the emission point, while they become much smaller further downwind from the plume as the mixing effectively reduces spatial gradients in the tracer field. To avoid any potential disturbances due to these numeric effects, and also to avoid representation errors due to insufficient spatial resolution in the near field, we have excluded data from the first two kilometres downwind of BPP from the analysis. We've found the model mass conservation scheme works well, with the total mass of both plume versions agreeing

within 0.035 % at distances between 2 km to 22 km, thus we treat both realisations of the plume as identical. See sect. S4 in the supplement for details.

### 3.2   Inferring point source rate using cross-sectional estimates

Undulations are visible in the column-averaged tracer (Fig. 4), which in turn leads to significant variability in the apparent emission rate calculated across the downwind distances (Fig. 5). The predicted Φ values vary between 22.5 $Mt\,yr^{-1}$ and 70.0

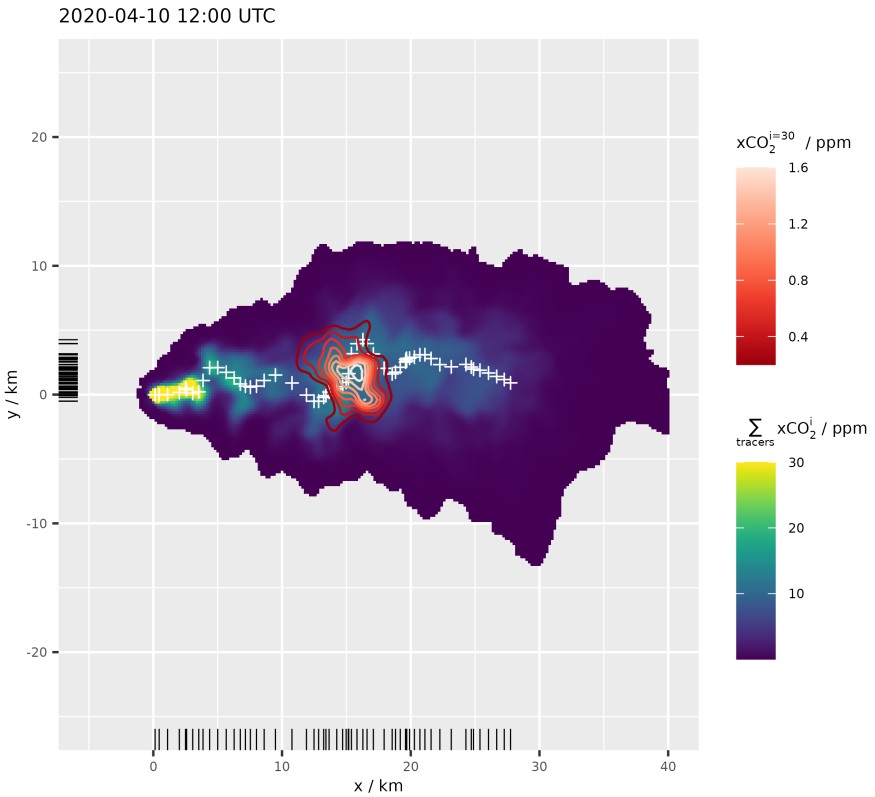

**Figure 4.** Simulated plume structure on 12:00 UTC 10 April 2020. $XCO_2$ of the full $CO_2$ plume (blue-yellow scale) fragment, with the distribution of a single puff (co2_bpp_30) shown in red-white contours (scaled by mole fraction). Values lower than 0.1 ppm were omitted for clarity. Centroid positions $[\overline{x_p}, \overline{y_p}]$ of puffs are marked with white crosses, and rug marks on the X and Y axes show their respective distributions.

$Mt \, yr^{-1}$, respectively 59 % and 182 % of the actual emission rate of $38.4 \, Mt \, yr^{-1}$. While the oldest puffs have been advected to over 30 km downwind from the source, the range over which they are equal to the full-signal tracer is only identical up to approximately $x = 22$ km. Beyond that distance, a steadily increasing fraction of the crosswind-aggregated signal comes from $CO_2$ emitted before 09:00 UTC. Therefore, we focus on values at downwind distances between $2 \, \text{km} < x < 22 \, \text{km}$ for the subsequent quantitative analysis.

Using the simulated fields, we estimated the mean emission from the source as an average of $\Phi(x)$ values at individual cross-sections, yielding $45.4 \, Mt \, yr^{-1}$. We then calculated the autocorrelation function of the apparent emission (Fig. 6) to estimate the $n_{\text{eff}}$. We assumed that the ACF overshoot for values above 4 km is primarily due to moderate sample size and can be ignored. Therefore for the $n_{\text{eff}}$ calculation, we set the terms beyond the first zero-crossing (at approximately $x = 4$ km) to zero, yielding $n_{\text{eff}}$ equal to 5.56, corresponding to an independent measurement occurring every 3.6 km. The calculated 1-$\sigma$





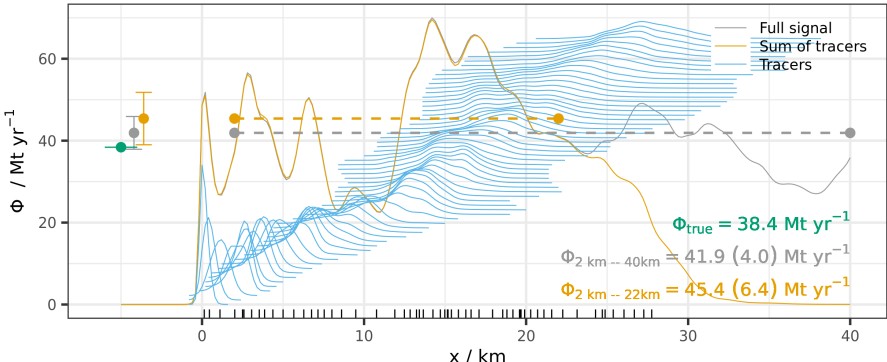

**Figure 5.** Apparent emissions $\Phi(x)$ downwind from the emission point calculated using the full simulated plume (grey) and the sum of the tagged tracers (orange). Contributions from individual tagged tracers are marked in blue (offset by a constant value for display purposes). Horizontal dashed lines mark the distance ranges over which the average emissions were estimated. Points to the left of the emission point compare true emissions in the model against computed averages with their uncertainty.

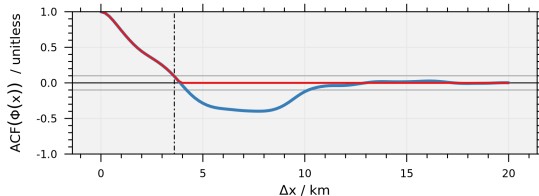

**Figure 6.** Blue: ACF of $\Phi(x)$ (blue), calculated for $2\,\mathrm{km} \leq x \leq 22\,\mathrm{km}$. Red: simplified ACF used for calculating $n_{\mathrm{eff}}$. The vertical dashed line denotes the distance between independent observations ($d_{\mathrm{indep}}$) corresponding with the calculated $n_{\mathrm{eff}}$

uncertainty of the mean emission is equal to $6.4\,\mathrm{Mt\,yr^{-1}}$ (14.2 % of the mean), thus the true emission value of $38.4\,\mathrm{Mt\,yr^{-1}}$ falls outside of the calculated 1-$\sigma$ range by $0.6\,\mathrm{Mt\,yr^{-1}}$.

Using the full-tracer signal rather than the sum of tagged tracers between $2\,\mathrm{km}$ and $40\,\mathrm{km}$, it was also possible to test whether increasing the distance, over which apparent emissions are estimated improves the precision of the emission estimates. Indeed, once this additional signal is included, the mean estimated emission yielded at $41.9 \pm 4.0\,\mathrm{Mt\,yr^{-1}}$ (relative uncertainty of 9.6
%). Results are summarized in Tab. 1.

When analyzing the spatial distribution of plume centroids (white crosses in Fig. 4), a meandering pattern is visible, caused by high-frequency variability of the wind fields downwind from the emission source. This meandering results in an uneven distribution of the centroids along both the X and Y axes (rug marks, Fig. 4). In order to quantify the effect on apparent emission estimates at given downstream locations, the normalized anomaly of the puff density $\lambda_c$ has been calculated as described in
sect. 2.8 and is shown in Fig. 7A. The centroid density anomaly is positively correlated with the corresponding apparent emission estimates ($\mathrm{R}^2 = 0.54$, Fig. 7B).





**Table 1.** Estimated emission statistics.

| Analysed x range | $n$ | $n_{\text{eff}}$ | $d_{\text{indep}}$* | $\overline{\Phi}$ | $u(\overline{\Phi})$ | $u_r(\overline{\Phi})$ | $u_a(\overline{\Phi})$ | $u_{ra}(\overline{\Phi})$ |
|---|---|---|---|---|---|---|---|---|
| Units | | | km | $\text{Mt yr}^{-1}$ | $\text{Mt yr}^{-1}$ | % | $\text{Mt yr}^{-1}$ | % |
| 2 km – 22 km | 101 | 5.6 | 3.6 | 45.4 | 1.4 | 3.0 | 6.4 | 14.2 |
| 2 km – 40 km | 191 | 9.2 | 4.1 | 41.9 | 0.8 | 2.0 | 4.0 | 9.6 |

\* $d_{\text{indep}}$ denotes the distance between independent observations.

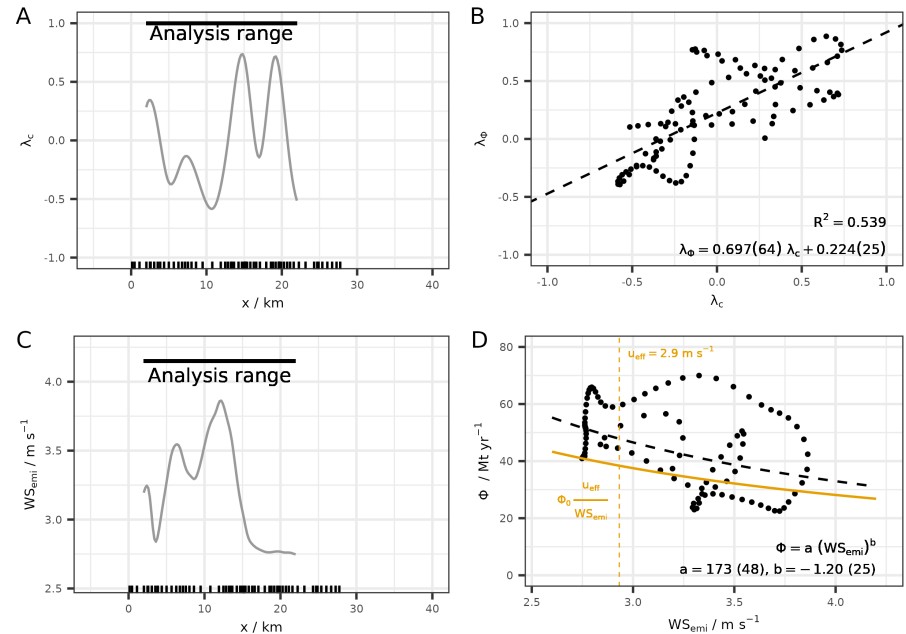

**Figure 7.** Panel A: Normalized anomaly of puff centroids ($\lambda_c$) per distance. Panel B: correlation of normalized anomaly of apparent emission ($\lambda_\Phi$) with $\lambda_c$. Panel C: Weighted average of parallel wind component at emission time ($\text{WS}_{\text{emi}}$) vs. distance. Panel D: Correlation between $\text{WS}_{\text{emi}}$ and $\Phi(x)$, together with the theoretical curve and fitted hyperbolic curve.

We have also analysed the correlation between the apparent estimated emissions and $\text{WS}_{\text{emi}}$ (shown separately as a function of $x$ in Fig. 7C). $\Phi(x)$ is observed to decrease with $\text{WS}_{\text{emi}}$, at an average rate of 18.1 $\text{Mt yr}^{-1}$ per every 1 $\text{m s}^{-1}$ of wind (Fig. 7D), however, the correlation of a linear fit is weak ($R^2 = 0.20$).

In order to understand how the dilution of $CO_2$ at the emission point due to local wind variability affects the apparent emissions, it is worthwhile to consider a simplified 1-D theoretical model of the relationship between the local horizontal wind at the emission point and the apparent emissions. Given constant emissions ($\Phi_0$) and effective wind speed ($u_{\text{eff}}$), the deduced emissions downwind (which are proportional to the downwind concentration enhancement, as per Eq. (5)) should be inversely





proportional to the instantaneous (turbulent) wind speed at the time of the emissions ($WS_{emi}$), i.e.:

$$\Phi \propto \Phi_0 \frac{u_{eff}}{WS_{emi}} \tag{13}$$

We have added the theoretical curve following Eq. (13), assuming that the proportionality factor equals exactly 1, as the yellow line in panel 7D, using $u_{eff} = 2.9 \text{ m s}^{-1}$) (value at 12:00). We have also added a non-linear least squares regression to fit a power curve ($\Phi = a(WS_{emi})^b$) to the data, yielding exponent $b$ equal to $-1.20 \pm 0.25$.

## 4 Discussion

Our model setup captured the expected characteristics of the point source plume structure well. The estimated cross-section emissions show typical features of the pollutant plume in terms of horizontal and vertical dispersion. Virtually all of the $CO_2$ plume is contained within 10 km from the main wind axis, and most of the mass is concentrated within 5 km, similar to the extent observed by OCO-3 and reported earlier (Figs 1, 3 & 4). It is likely that vertical mixing is overestimated in the direct vicinity of the emission point, which is at least partially caused by the use of a Gaussian emission profile rather than having the plume rise mechanism implemented directly in the model. This potential inaccuracy becomes less relevant with distance, as vertical mixing efficiently distributes the tracer throughout the PBL. The variability of the apparent emissions predicted by the model are remarkably similar to those based on remote sensing observations from the same day, with a modelled $1$-$\sigma_\Phi$ of $11.6 \text{ Mt yr}^{-1}$ calculated for distances between 2 km and 40 km vs. $11.0 \text{ Mt yr}^{-1}$ observed by Fuentes Andrade et al. (2024, pink range marked as SD in Fig. 1).

The number of independent observations appears to be in good agreement as well, with our modelling framework predicting an independent $\Phi$ estimate every 3.6 km compared to one every 2.9 km estimated by Fuentes Andrade et al. (2024) using a slightly different approach. Based on the above we conclude that the overall plume structure is realistic, which gives us confidence that the tagged tracers propagated via the model also realistically depict the distribution of the tracer mass. This is an important conclusion, as this assumption is virtually impossible to test directly in the field.

In order to estimate the influence of the turbulence on the precision of emission estimation, we have calculated the mean apparent flux for two plume segments, namely for the full available plume distance, using the full tracer (2–40 km) and a shorter one (2–22 km), corresponding to the section of plume fully resolved by the tagged tracers. In both instances, we have estimated the mean emission uncertainty following the algorithm provided in Sect. 2.7. When the correlation of observations is taken into account, the uncertainties of the emission estimate are significantly higher, increasing by a factor of four relative to the actual emissions for uncorrelated uncertainty (Table 1). This extra uncertainty stems purely from turbulent dispersion and will always affect observations made under turbulent conditions.

It is not straightforward to compare the obtained uncertainty estimates against the literature, as methodologies of uncertainty estimation vary widely across studies. In a recent publication focused on estimating BPP emissions using OCO-3 data using GPI method, Nassar et al. (2022) reported a range of total uncertainties between 4.1–19.9 % (mean of 12.3 % over 10 analysed cases), but identified that the largest uncertainty stemmed from either background estimation (in 60 % of cases) or wind speed





(40 % of cases), neglecting the correlation in the observational data altogether. In another study (Fuentes Andrade et al., 2024), a total uncertainty of 6.38 $\mathrm{Mt\,yr^{-1}}$ was reported for an OCO-3 scene from 10 April 2020. The contribution of dispersion on the uncertainty estimate was calculated explicitly and found to result in a 10.5 % relative uncertainty on the BPP emission estimate, consistent with the value reported here.

Moreover, over the nine scenes reported in that study (all collected from April–October, when convective activity is common), the relative uncertainty due to dispersion was found to be between 7.4 % to 22.0 % of the total emission, with an average of 14.9 %. For the same subset Fuentes Andrade et al. have estimated an average total relative emission uncertainty of 22 %, underlining the importance of turbulence in the overall emission uncertainty.

Based on the literature and the results of our current study, we conclude that the presence of turbulence provides a lower
bound to the precision of source estimation that cannot be overcome when using the CSF method, irrespective of whether it is applied to spaceborne or airborne measurements. The relative contribution of this error is expected to be smaller under conditions with weaker turbulence, however, this causes practical difficulties as these usually occur in situations suboptimal for satellite remote sensing retrievals via passive instruments (e.g. nighttime, winter, cloud cover). As the spatial correlation of the signal reduces the effective number of measurements, it is clear that the turbulence will also negatively impact the precision
of emission estimates using other methods, like GPI, while those that attempt to integrate the signal over larger distances (e.g. IME) would likely avoid it to a certain degree. A detailed investigation of the effect of turbulence on the precision of these methods, however, falls outside of the scope of the current study but is worth considering in the future.

In the case of airborne measurements, the consequences of the correlation of $\Phi$ in CSF on estimation uncertainty can be even larger, as generally much fewer observations are available. For example, during one of the flights during the CoMet 1.0
campaign, only sixteen in-situ downwind cross-wind tracks were executed, two by HALO (a German research aircraft) and fourteen by a smaller Cessna aircraft operated by DLR (Gałkowski et al., 2021; Brunner et al., 2023). In general, a certain sampling density is needed to be able to estimate the scale of correlation (i.e. the number of independent observations) or inflated uncertainties related to turbulence would need to be assumed.

Using tagged tracers, we were able to study closely the mechanics of the plume dispersion. Based on the simulation results,
the imprint of turbulence on the emitted plume in a turbulent PBL starts close to or even at the stack, where the tracer is extremely localised. In general, three mechanisms occurring near or at the emission location can create a variable structure in the tracer mole fraction fields like the one observed.

The first is the uneven vertical distribution caused by vertically variable atmospheric advection. The extreme case of the effect would occur when updrafts elevate most of the emitted puff close to the PBL top with the simultaneous occurrence
of a strong vertical wind gradient, effectively transporting the affected puff quicker than others for a limited time and likely also altering its direction. No strong evidence is found for this on this day in our model, however - while some wind shear was indeed observed in the simulated tracer distributions, this effect is dampened in our simulation due to the relatively large vertical extent over which the plume is injected into the model.

The second mechanism is related to variability in the horizontal wind speed at the emission point due to the occurrence of
larger eddies. Variations of the wind speed and direction associated with such eddies cause dilution or enrichment relative to



the average, depending on whether the local wind speed is higher or lower than the $u_{\text{eff}}$. Thanks to the simulation of puffs we were able to investigate the influence of variability in $\text{WS}_{\text{emi}}$ (horizontal wind speed at emission source, parallel to the X axis) on the resulting plume. If the dilution at the source was the only mechanism responsible for the observed variability, the relationship between the $\Phi$ downwind and the wind speed at the emission point is given in Eq. (13) and plotted in yellow in Fig. 7D). The actual spread over the calculated tagged tracers is much higher, reflecting previously discussed complexities in the realistic turbulent flow. Nevertheless, the mean relationship does show a decrease of apparent emission with increasing local wind speed, $\text{WS}_{\text{emi}}$, with the non-least square regression suggesting some proportionality to the inverse of $\text{WS}_{\text{emi}}$, albeit it is clear that a simple proportionality to horizontal winds is insufficient to explain the relationship.

The third potential source of variability is the coherent transport of the tracer mass in directions perpendicular to the mean advection (X axis), which can occur downwind from the emission point in the presence of large eddies. While it is unlikely that such coherent across-wind mass transfers play a significant role at larger distances (where the characteristic turbulent scale causes only random movements), we observe such movements close to the emission point, where a significant portion of the tracer mass can be transported in the y-direction by individual eddy structures. This causes a meandering effect, which can in some cases increase the density of the tracer at a given distance x (and thus add to variability in $\Phi$), as seen close to $x = 15\text{km}$ (c.f. Fig. 4 & 5).

By following the centres of mass (centroids) of each tagged tracer, we were able to determine that the relationship between the estimated emission $\Phi$ and the density of the mass centroids is approximately linear (Fig. 7 A & B, $R^2 = 0.539$). A positive correlation is expected, as the increase in density of centroids represents the increased density of tracer mass per unit distance along the X axis. Departures from the linearity in this relationship might be attributed to a) variability of wind (speed and direction) during the 3-minute release time of each puff causing additional apparent diffusion, or b) potential spatial gradients in the wind field in the area downwind of the plume (e.g. due to divergence or convergence at larger scales), rendering the assumption of a constant $u_{\text{eff}}$ in Eq. 5 invalid.

## 5  Conclusions

Our study corroborates the critical role of turbulence in estimating emissions from plume observations.

We applied a realistic model setup to simulate a typical turbulent plume emitted from a power plant, and have shown that coherent spatial structures in the plume are formed at and near the emission point and persist across relatively large downwind distances, likely over $30\,\text{km}$ (the distance over which we studied the effect). We then applied a commonly used cross-sectional flux technique to infer the emission rate of a point source, mimicking the error-free retrieval of a remote sensing imager of sub-kilometre-scale resolution. We have found that in the turbulent atmosphere, even for an idealized case of observing a strong plume structure emitted from a known point source with perfectly known background distribution and wind speed, the uncertainty of the estimated emissions is limited by the variability caused by large-scale eddies present in the atmospheric flow. In the analysed case this uncertainty was estimated to be 14.3 %, in line with previously reported contributions from dispersion uncertainty (Fuentes Andrade et al., 2024). When applied to actual observational data, this uncertainty can only



be increased, primarily due to imperfect knowledge of the wind fields and inaccuracies in the background estimation. In this
study, the conclusion has been drawn for the application of the cross-sectional flux method for an idealized remote sensing
instrument, however, the conclusions are valid for other methods, as the physics causing the observed signal variability will
still be present. Correlation of the observed signal that reduces the number of effective observations is of particular importance
here.

It should be noted that the effects of the persistent turbulent structures are likely less severe for a) weaker turbulence regimes,
and b) when the spatial scales of the emissions become comparable to the spatial scales of the eddies present in the atmosphere,
preventing the formation of coherent structures in the downwind signal. Thus estimations of point sources, like the one dis-
cussed (power plant stack), are affected to a larger degree than e.g. megacities that spread the emissions over larger areas.

We have attempted to isolate the primary causes of the observed variability in the downwind structure of the plume. By using
temporally-tagged tracers, we have managed to relate the variability of the downwind structures in the distribution of tracer
mole fractions to the variability in the wind field at the emission point and found indications that at least part of the observed
variability can be related to the initial dilution of the tracer into the atmospheric flow along the main wind direction. The
relationship between the parallel wind speed at emission and the resultant emission estimate is not straightforward, reflecting
the stochastic nature of turbulent motions within the plume.

Overall, we believe that the results of the study highlight challenges that emission estimation using modern observational
methods will face in the future. This is directly related to turbulent motion in the atmosphere, which cannot be removed or
corrected. The instantaneous (turbulent) winds at or near the point source (at the height corresponding to the effective emission
height, including plume rise for power plants) are chaotic in nature and cannot be predicted. While it is theoretically possible
to observe them at sufficient temporal resolution and within the necessary vertical extent (e.g. using 3D wind lidars), the fact
that they are only weakly correlated to the downwind plume structures makes it necessary for the impact of turbulence to be
treated as a stochastic effect. Due to its influence on the uncertainty of emission estimates, it needs to be considered both in
the currently available methods, as well as in the design of future satellite and airborne capabilities targeted at point source
emission estimation.

*Data availability.* The model configuration files and production scripts for model runs will be be made available through a public repository
upon publication. Due to significant amount of disk space required, the model results will be archived on DKRZ HSM data storage[2] upon
publication, and will be accessible per request.

*Author contributions.* MG and CG conceptualized the study. MG performed model runs, processed the data and wrote the paper. BFA and JM
provided input and corrections to sections related to remote sensing methods and analysis. JM and CG advised and contributed significantly
to experimental design and analysis. All authors contributed to the interpretation of the results and to the writing of the article.

[2]https://docs.dkrz.de/doc/datastorage/hsm/index.html



*Competing interests.* At least one of the (co-)authors is a member of the editorial board of Atmospheric Chemistry and Physics.

*Acknowledgements.* This work is funded by the German Federal Ministry of Education and Research (BMBF) project "Integrated Green-house Gas Monitoring System for Germany – Modellierung (ITMS M)" under grant number 01 LK2102A, and by "ITMS - Modul Beobach-tung II - Verbundprojekt Hochaufgelöste Satellitenbilder von XCO2 und XCH4 - Teilprojekt 2: Simulationen" under grant number 01LK2309B. This work also used resources of the Deutsches Klimarechenzentrum (DKRZ) granted by its Scientific Steering Committee (WLA) under project ID bm1400. The authors would also like to thank Mark Schlutow (MPI-BGC) for his valuable input.



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
