# Peer review of "Impact of atmospheric turbulence on the accuracy of point source emission estimates using satellite imagery"

_EGUsphere, 2024_

## Referee Comment (RC1)

**Review: *Gałkowski et al. Impact of atmospheric turbulence on the accuracy of point source emission estimates using satellite imagery**

**Summary**

Gałkowski et al. present the effect of atmospheric turbulence on $CO_2$ emission rate estimates They apply a high-resolution simulation with temporally tagged tracers and discuss the decomposition of the plume variability. This is novel and useful for analyzing high-resolution satellite data. However, the study only shows one timestamp result, and this point needs to be addressed in a revision. Finally, there are a number of minor technical corrections that are needed for readability.

**General comments**

1. The authors pick the time between 9 April 2020 and 10 April 2020 for the simulation. Since the turbulence is the key point, it is better to discuss the variation of wind speed and direction before further analysis.

2. It makes sense to only pick the simulation at noon, which is "consistent with typical observation times of passive remote sensing". As mentioned above, the wind speed may limit the application. More simulations with different wind speed would make the conclusion more concrete. The authors may aim to compare the result with Fuentes Andrade et al. (2024), if so, I would suggest picking more OCO-3 observations and running the simulations.

3. Since the turbulence effect is important for estimate uncertainty, could authors define a new effective wind (Ueff) based on simulation and apply it to correct emission rates derived from satellite observations?

**Specific comments and technical corrections**

- L8: "However, a realistic evaluation of the accuracy and precision of the obtained estimates is essential."

- L12: ".... significant uncertainties ... on the order of 10 % of the total source strength, in retrieved ..."

- L21: "The ongoing warming is already driving widespread adverse impacts across various components of the Earth's system, not only degrading the environment but also directly affecting communities by intensifying extreme events such as heatwaves, heavy precipitation, droughts, and tropical cyclones." And please add the references.

- L23: "Furthermore, if left unmitigated, the consequences of elevated $CO_2$ levels will persist for centuries."

- L31: Please add the references.

- L35: "Independent, science-based observations of GHG emissions offer a promising approach to enhancing the confidence of all stakeholders."

- L37: "Moreover, the monitoring of atmospheric greenhouse gases through a combination of in situ and satellite-based measurements, integrated with modern top-down frameworks, can  reduce uncertainties in global anthropogenic flux estimates, which averaged 9.6 ± 0.5 GtC yr$^{-1}$ over the past decade."

- L43: What is the definition of "currently available tool"? Statistic model or satellite observations?

- L48: Please add the reference for "... strongest emissions occur (US, EU, China, India)."

- L62: "Satellite observations offer a distinct advantage due to their global coverage and lower cost per observation."

- L73: "(GPI, Krings et al., 2011; Nassar et al. (2017)"

- L75: "... has been widely used in practical applications in recent years."

- L77: Missing the definition of "estimation statistics".

- L81: What's the wind speed range for the "10~20% uncertainty"? Is the wind speed the most important one in these parameters (*the estimation method employed, wind speed, stability conditions, time of day*)?

- L85: How are the data collected virtually?

- L89: Figure 1 is copied from another paper and too small. It would be better to apply the CSF method and make a new plot. Or just remove it.

- L93: "We employ high-resolution WRF-GHG simulations over a previously studied point source, enhancing the modeling system with temporally tagged tracers."

- L100: "The Bełchatów Power Plant (BPP) is one of the largest anthropogenic $CO_2$ point sources globally, relying on lignite coal for power generation."

- L103: "Under both national and EU legislation, accurate information on GHG emissions and operational status is publicly accessible, making the Bełchatów Power Plant an ideal target for developing and testing new instruments and methods."

- L116: Please introduce the WRF model briefly.

- L126: " .... (ECMWF, 2022)"

- L169: How did the authors define the segmentation? Will that affect the turbulence analysis?

- L170: "The resulting $CO_2$ signals are conceptually similar to the "particles" or "air parcels" used in Lagrangian models."

- L171: Adding the illustration of puff to the main text could help readers better understand it. I would suggest plotting three puffs with same time steps to show the transport and turbulence. As the authors discuss the difference between tracer and the sum of puffs around L310, it is also useful to include that in this figure.

- L173: As mentioned in the "General comments", analysis with more cases would be better.

- L202: Do the lower and upper integration limits of y depend on the segmentation described in L169?

- L207 and L227: The definition of Ueff at L227 is different from Varon's paper: "*U*eff in the cross-sectional flux method is different than *U*eff in the IME method. For each plume in the training set, *U*eff is computed from Eq. (6) based on *C* and the known source rate *Q*.". Please clarify the Ueff definition.

- L233: How about normalizing the data by the true emission rates?

- L237: What is the definition of "typeB uncertainty"?

- L241: It would be better to explain the ACF in detail.

- L265: The sentence is duplicated with L263—L264.

- L301: "As shown, the model …"

- L337: The true emission value is within the full-tracer estimates. It would make readers feel that this method is better. Is this conclusion still true for other cases? Another point: Is the full-tracer method standing for the application of CSF to real satellite observation? It is better to mention the relationship between simulation estimate and real observation.

- L355: As mentioned before, please clarify how the authors calculate the Ueff values.

- L357: Correct the format: "using ueff = 2.9 ms−1) (value at 12:00)."

- L358: Which regression is better?

- L368: I could not find the source of "11.6 Mt yr-1". Please correct me if I missed.

- L380: Please add the definition to Table 1. It is difficult to find out which variable is *"emissions for uncorrelated uncertainty"*.

- L383: The comparison makes me curious how to apply author's method to real satellite data? or does the comparison mean the dispersion uncertainty in *Fuentes Andrade et al., (2024)* is accurate enough?

- L390: What is the reference of *"in that study"*?

- L400: Does this mean that the turbulence will cause a larger uncertainty for GPI and IME method? Please make it clearer. It is valuable to apply the IME

with different limits (e.g. 2-22 and 2-40 km) and check the differences in the supplement.

- L444-445: Please combine them into one paragraph.

- L454: Why does the real uncertainty can only increase? As the effective wind speed aims to minimize the difference between estimates and true emission rates, the turbulence effect can be included there.

---

## Author Response (AR1)

**Response to reviewers**

What follows is the response to two reviews to the manuscript submitted to EGUsphere titled:

"Impact of atmospheric turbulence on the accuracy of point source emission estimates using satellite imagery", by Galkowski et al., available at:

https://egusphere.copernicus.org/preprints/2024/egusphere-2024-2792/

Before addressing specific comments, we would like to offer our sincere thanks to the reviewers for the time and attention given to our manuscript, and for all the suggested corrections, both scientific and language-related. All are highly appreciated.

The original text from the reviewer is given in black, while the responses from authors are in blue. Text cited from the manuscript is in *italics*. In this document, we refer to the original manuscript using Lx notation (consistent with the reviewer's comments), while an R index is added when the revised manuscript is cited. For example, L15 refers to the original manuscript, while RL15 refers to the revised version.

**Response to Review #1**

**Summary**

Gałkowski et al. present the effect of atmospheric turbulence on CO2 emission rate estimates They apply a high-resolution simulation with temporally tagged tracers and discuss the decomposition of the plume variability. This is novel and useful for analyzing high-resolution satellite data. However, the study only shows one timestamp result, and this point needs to be addressed in a revision. Finally, there are a number of minor technical corrections that are needed for readability.

We thank the reviewer for appreciating the topic's importance. Please find our detailed responses below.

**General comments**

1. The authors pick the time between 9 April 2020 and 10 April 2020 for the simulation. Since the turbulence is the key point, it is better to discuss the variation of wind speed and direction before further analysis.

We have followed the reviewer's suggestion and added a new subsection (3.1.) to the Results, where we discuss the wind conditions and the characteristic eddy sizes based on simulation results. We also added a new figure that shows the wind speed and direction variability in the area of interest. This can be found at RL329.

2. It makes sense to only pick the simulation at noon, which is "consistent with typical observation times of passive remote sensing". As mentioned above, the wind speed may limit the application. More simulations with different wind speed would make the conclusion more concrete. The authors may aim to compare the result with Fuentes Andrade et al. (2024), if so, I would suggest picking more OCO-3 observations and running the simulations.

We thank the reviewer for this suggestion. We agree that the results from runs simulating a larger range of wind speeds would provide more robust estimates of the effect. However, the main aim of this paper is to establish the basic mechanism leading to the spatial variations. Running simulations for the cases discussed by Fuentes Andrade et al. (2024) is prohibitively expensive, and beyond the scope of the current study.

Focusing on one example case allows us to focus on and explain the transport mechanism leading to the variability on the spatial scales seen in Fuentes Andrade et al. (2024), which, to our knowledge, is a new contribution to the literature. We are considering the simulation and analysis of additional plumes under different meteorological conditions in the context of future studies.

3. Since the turbulence effect is important for estimate uncertainty, could authors define a new effective wind (Ueff) based on simulation and apply it to correct emission rates derived from satellite observations?

We are not sure if we understand this comment correctly. A correction of the impact of turbulence is not really possible as the process is stochastic, i.e. the realization of turbulent

eddies in the simulation do not match those observed in reality. We therefore would not think such a method to succeed.

While we could follow the approach of Varon et al. (2018), the robustness of the obtained relationship between our ueff and emissions would be limited, as we only analyze a single case. It is insufficient to determine a formula applicable to a wide range of conditions expected in reality.

**Specific comments and technical corrections**

L8: "However, a realistic evaluation of the accuracy and precision of the obtained estimates is essential." – Agreed.

L12: ".... significant uncertainties ... on the order of 10 % of the total source strength, in retrieved ..." – we altered the sentence, it now reads:

We demonstrate that persistent structures in the downwind concentration fields of emitted plumes can cause significant uncertainties in the retrieved fluxes on the order of 10 \% of the total source strength, when the commonly used cross-sectional mass-flux (CSF) method is applied with short distances between individual estimates.

L21: "The ongoing warming is already driving widespread adverse impacts across various components of the Earth's system, not only degrading the environment but also directly affecting communities by intensifying extreme events such as heatwaves, heavy precipitation, droughts, and tropical cyclones." And please add the references.

Thank you. We have decided to discard this fragment of the text following the suggestion by Reviewer 2. See below.

L23: "Furthermore, if left unmitigated, the consequences of elevated CO2 levels will persist for centuries."

As above.

L31: Please add the references.

Added reference to the "2006 IPCC guidelines for national greenhouse gas inventories", Eggleston et al., 2006, available at: https://www.ipcc-nggip.iges.or.jp/public/2006gl/

L35: "Independent, science-based observations of GHG emissions offer a promising approach to enhancing the confidence of all stakeholders." – changed as suggested

L37: "Moreover, the monitoring of atmospheric greenhouse gases through a combination of in situ and satellite-based measurements, integrated with modern top-down frameworks, can reduce uncertainties in global anthropogenic flux estimates, which averaged  $9.6 \pm 0.5$  GtC yr-1 over the past decade."

The sentence that this comment concerns was discarded following a suggestion from Reviewer #2. See below.

L43: What is the definition of "currently available tool"? Statistic model or satellite observations?

As above, this fragment was discarded. We meant mainly the inverse frameworks utilizing surface observations.

L48: Please add the reference for "... strongest emissions occur (US, EU, China, India)."

Added reference to EDGAR (Emissions Database for Global Atmospheric Research; Janssens-Maenhout et al. 2019) and listed the entities alphabetically.

Janssens-Maenhout, G., Crippa, M., Guizzardi, D., Muntean, M., Schaaf, E., Dentener, F., Bergamaschi, P., Pagliari, V., Olivier, J. G. J., Peters, J. A. H. W., van Aardenne, J. A., Monni, S., Doering, U., Petrescu, A. M. R., Solazzo, E., and Oreggioni, G. D.: EDGAR v4.3.2 Global Atlas of the three major greenhouse gas emissions for the period 1970–2012, Earth System Science Data, 11, 959–1002, https://doi.org/10.5194/essd-11-959-2019, 2019

L62: "Satellite observations offer a distinct advantage due to their global coverage and lower cost per observation." – Changed as suggested. See also response to the related comment from Reviewer #2 below.

L73: "(GPI, Krings et al., 2011; Nassar et al. (2017)" – corrected

L75: "... has been widely used in practical applications in recent years."

A comment regarding L75 was also made by Reviewer #2. See below.

We have altered the fragment these comments concern for better readability and text flow, with the suggested changes (by both reviewers) taken into the account.

L77: Missing the definition of "estimation statistics". – revised this sentence, it now reads:

Current developments include improvements of plume detection algorithms, (...), robust statistical analyses of emission estimates from repeated scenes by a single spaceborne instrument (Nassar et al., 2022; Fuentes Andrade et al., 2024; Santaren et al., 2025), and using detailed bottom-up information for comparisons (Nassar et al., 2022, Fuentes Andrade et al., 2024).

L81: What's the wind speed range for the "10~20% uncertainty"? Is the wind speed the most important one in these parameters (the estimation method employed, wind speed, stability conditions, time of day)?

A wide range of wind speeds has been reported across the cited studies, usually between 2 m/s and 10 m/s. This range is consistent in cited studies that used synthetic data experiments, e.g. Varon et al. (2018) and Santaren et al. (2025; new reference suggested by Reviewer 2). Similar ranges were reported in papers using actual observations (Nassar et al. 2022, Fuentes Andrade et al. 2024).

With regard to the second question, the range of uncertainty given here is meant as a general order of magnitude to be expected under typical conditions for which satellite images are

available – close to midday on sunny days characterized by neutral or unstable conditions, with limited cloud cover.

The uncertainty in estimating wind speed is usually reported as one of the most important uncertainty sources, however there is no consensus between studies as to how exactly to quantify the error components. Fuentes Andrade (2024) reported between 24% to 82% of the variance coming from the wind, similar to Nassar et al. (2022) – between 10% to 84%, however with very different estimates for the same scenes. Kuhlman et al. (2021) stated that "wind speed uncertainty accounts for less than 5% of the total uncertainty", but subsequently added that some part of uncertainty related to wind might have been counted in other components. Other uncertainty components recognized as significant included the instrument precision (Varon et al., 2018; Kuhlmann et al., 2019), background estimation (Kuhlmann et al., 2019, 2020), plume rise (Nassar et al., 2022) and others, e.g. "method error" in Kuhlmann et al. (2020) and "sensitivity uncertainty" in Fuentes Andrade et al. (2024).

We have reworked the paragraphs discussing uncertainties in the introduction (L81-L89) and expanded the discussion to include the information given above. It can be found in RL73—RL84. We have also included information about the wind speed ranges reported in the cited literature, as requested.

L85: How are the data collected virtually?

"Virtually all" is used here as a substitute of "almost all". Left without change.

L89: Figure 1 is copied from another paper and too small. It would be better to apply the CSF method and make a new plot. Or just remove it.

We would like to stress that the figure in question was appropriately cited and taken from a publication from one of the coauthors of this study. Our intention to include it here was to allow the readers easy access to the previously published results, but we agree that its quality was insufficient.

We have removed the figure and added a reference to the source publication, as suggested.

L93: "We employ high-resolution WRF-GHG simulations over a previously studied point source, enhancing the modeling system with temporally tagged tracers." -- implemented as suggested.

L100: "The Belchatów Power Plant (BPP) is one of the largest anthropogenic CO2 point sources globally, relying on lignite coal for power generation." – Changed as suggested

L103: "Under both national and EU legislation, accurate information on GHG emissions and operational status is publicly accessible, making the Belchatów Power Plant an ideal target for developing and testing new instruments and methods." – Changed as suggested

L116: Please introduce the WRF model briefly.

We would like to point out that the whole paragraph in L114-L120 serves as such a brief introduction, with information about the model purpose, history and general structure,

supported by the primary literature source – a commonly cited publication by Skamarock et al., 2008.

L126: ".... (ECMWF, 2022)" – thank you for spotting this. Changed.

L169: How did the authors define the segmentation? Will that affect the turbulence analysis?

To better explain the segmentation and why we chose a three-minute averaging time, we have included a new paragraph after L167 (RL177-RL182). Regarding the second part of the question, we compared the one-minute instantaneous wind speeds with three-minute averages, as shown in Figure 1 below. We compared the variance of the wind speed at the emission point and found only a minor loss of 1.5 % of variance when comparing 3-minute averages against 1-minute instantaneous values (variances of 0.333 m²/s² and 0.328 m²/s², respectively). Thus, shorter emission segments would not significantly alter the results. The simulation cost would increase significantly, however.

Figure 1: Wind speeds (ws, in m/s) at the emission point as a function of time of day, comparing instantaneous 1-minute wind speeds (in pink) and three-minute averages (in cyan).

**The added text reads:**

We also used 60 additional tracers tagged by the time of release (temporally-tagged tracers) in order to study the effects of atmospheric turbulence on source estimation inference. Each tracer corresponds to a short segment of the emitted plume, together encompassing the full emission signal emitted from the stack over a three-hour time period (see the next section). Three-minute segments were chosen as a compromise between the desire for maximum detail and computational constraints. This time was sufficient to represent wind variability at the emission point. The numerical tests have shown only a 1.5% loss of variance when using three-minute averaged output as compared to instantaneous one-minute output.

L170: "The resulting CO2 signals are conceptually similar to the "particles" or "air parcels" used in Lagrangian models." – changed as suggested.

L171: Adding the illustration of puff to the main text could help readers better understand it. I would suggest plotting three puffs with same time steps to show the transport and

turbulence. As the authors discuss the difference between tracer and the sum of puffs around L310, it is also useful to include that in this figure.

We thank the reviewer for the suggestion. We would like to point out that a puff example is plotted against the sum of puffs in Figure 3 of the revised manuscript (formerly Figure 4), therefore we believe that adding yet additional plot is not needed.

We have expanded the text in L171 to point readers to that plot as well:

The distribution of  $CO_2$  mole fractions for a selection of puffs is presented in Fig. S2 of the supplementary material, and an example  $xCO_2$  from a single puff is plotted in Fig. 3 (see Section 3) against the full plume extent.

L173: As mentioned in the "General comments", analysis with more cases would be better.

Please see the response there.

L202: Do the lower and upper integration limits of y depend on the segmentation described in L169?

They do not. They only depend on the physical extent of the plume which is dependent on the atmospheric state on the given day. To put it simply, the shape of the plume remains the same regardless of what kind of segments we follow in the model.

In fact, apart from the segmented plume we have also simulated its classical realization with a single tracer. Both realizations were largely identical, with some minor numerical effects. See also discussion in L310-321, and supplement Section S3 (with Fig. S4).

L207 and L227: The definition of Ueff at L227 is different from Varon's paper: "Ueff in the cross-sectional flux method is different than Ueff in the IME method. For each plume in the training set, Ueff is computed from Eq. (6) based on C and the known source rate Q.". Please clarify the Ueff definition.

We thank the reviewer for this remark. Reviewer 2 also pointed this out, and we have altered the text to clarify the relationship between ueff as used in our paper and the definition from the Varon et al. (2018) study. Please see our comment to the remark from the second reviewer below.

**L233: How about normalizing the data by the true emission rates?**

Throughout the paper we try to look at the analysis from the measurement perspective, where the true emission rate is never known. We therefore used the average of the apparent emission rate.

**L237: What is the definition of "typeB uncertainty"?**

Thank you for pointing this out. This should in fact be Type A uncertainty, it has been corrected in the text.

Type A uncertainty is defined as the standard deviation of the mean of the random variable.

$$u(\bar{q}) = \sqrt{\frac{1}{n(n-1)} \sum_{j=1}^{n} (q_j - \bar{q})^2}$$

We followed the nomenclature from the "Evaluation of measurement data — Guide to the expression of uncertainty in measurement", JCGM 2008, (link: <a href="https://doi.org/10.59161/JCGM100-2008E">https://doi.org/10.59161/JCGM100-2008E</a>). This is consistent with the usage in the work of Zieba, 2010, which we apply.

We updated the text accordingly, also implementing changes suggested by Reviewer 2 (see below).

L241: It would be better to explain the ACF in detail.

We respectfully disagree, the autocorrelation function is a relatively standard statistical quantity well described in the reference provided, as well as multiple other books.

L265: The sentence is duplicated with L263—L264.

Thank you for spotting this. It has been deleted.

L301: "As shown, the model ..." – changed

L337: The true emission value is within the full-tracer estimates. It would make readers feel that this method is better. Is this conclusion still true for other cases? (...)

We would like to point out that the full-tracer estimate calculation is identical to the sum of puff-tracer estimates. Had we extended the number of puffs (in time) sufficiently, we would have been able to obtain the signal out to 40 km. We only used the full signal for the longer distances as the 180 minutes of emissions were responsible for the full plume only out to 22 km downwind. Beyond that distance the plume consists increasingly of the CO2 emitted prior to 09:00 UTC, i.e. before the first puff was emitted.

More generally, the CSF method relies on multiple cross sections to reduce uncertainty, so the longer the analyzed plume, the smaller the uncertainty of the estimate, as the persistent structures within the plume average out over distance. However, increased distance also means lower anomalies due to plume dispersion, with more of the downwind signal falling below the detection limit of a given instrument.

L337 (...) Another point: Is the full-tracer method standing for the application of CSF to real satellite observation? It is better to mention the relationship between simulation estimate and real observation.

With regard to the second point, we only applied the method to our simulated tracers. The application of the CSF method to real satellite observations was done in the study of Fuentes Andrade et al. (2024), to which we refer on several occasions. The relationship between their estimate and our modeled one is commented on in the Discussion section in L368-369.

L355: As mentioned before, please clarify how the authors calculate the Ueff values.

The description of the ueff calculation has been expanded. See comment to L207 by this reviewer (above) and to L208 by the second reviewer (below).

L357: Correct the format: "using ueff = 2.9 ms-1) (value at 12:00)." – Corrected

L358: Which regression is better?

We added the following comment to answer this question:

As expected, the empirical formula fits the data better. The difference between the theoretical curve is expected, as the assumptions for such a simple model are not fulfilled in a realistic three-dimensional case.

L368: I could not find the source of "11.6 Mt yr-1". Please correct me if I missed.

Thank you for pointing this out. Indeed, this number was a carryover from an earlier draft. We have now used a consistent number.

L380: Please add the definition to Table 1. It is difficult to find out which variable is "emissions for uncorrelated uncertainty".

The definitions were added as requested. We have also clarified the sentence by discarding "relative to the actual emissions for uncorrelated uncertainty".

L383: The comparison makes me curious how to apply author's method to real satellite data? or does the comparison mean the dispersion uncertainty in Fuentes Andrade et al., (2024) is accurate enough?

We did not apply our method to real satellite observations, as our focus was to demonstrate the general effect. This would require careful analysis of the error sources that we omitted. It could be attempted, as the cited literature provide a good starting point, but this is out of scope.

Fuentes Andrade et al. (2024) in fact applied a method similar to ours, and included the increase of uncertainty due to correlations in the extracted apparent emissions. Their "dispersion uncertainty" is directly comparable to our uncertainty of mean apparent emission. We would argue that our method is simpler in application.

The relationship is discussed in more detail in RL451-454.

In the study by Fuentes Andrade et al. (2024), a total uncertainty of 6.38 Mt yr-1 was reported for an OCO-3 scene from 10 April 2020. The contribution of dispersion on the uncertainty estimate was calculated explicitly and found to result in a 10.5 % relative uncertainty on the BPP emission estimate, consistent with 9.6 % obtained in our study (Table 1).

L390: What is the reference of "in that study"? – Fuentes Andrade et. al. 2024. Added.

L400: Does this mean that the turbulence will cause a larger uncertainty for

GPI and IME method? Please make it clearer. It is valuable to apply the IME with different limits (e.g. 2-22 and 2-40 km) and check the differences in the supplement.

We thank the reviewer for the suggestion, and would like to point out that applying IME is outside of scope of our study, however.

We have merged this remark with a comment from Reviewer 2 (see below), and rephrased the text in the following way):

As the spatial correlation of the signal reduces the effective number of measurements  $(n_{eff})$ , it is expected that the turbulence will also negatively impact the accuracy of emission estimates from other methods as well, especially when the estimates rely on observations collected close to the point source, where the spatial variability is higher. Because increasing the analysed distance reduces the total uncertainty in the CSF method, we anticipate that methods that rely on fitting large number of observations (like GPI or IMF) would be less affected, provided that sufficient data of downwind observations are available. In a paper recently published by Santaren et al. (2025), the authors analysed the performance of multiple estimation methods, including IME, GPI and CSF. The results showed that the CSF method generally outperformed the IME method. While the correlations of turbulent plumes were not taken into account, the uncertainty estimates are unlikely to be significantly biased, as the original 1 km x 1 km resolution of the simulations was further reduced to mimic CO2M satellite observations (to approximately 2-km spatial scale), with individual cross sections at distances of  $\sim$ 5 km to allow for enough data points for fitting. A detailed investigation on how the effect of turbulence affects the precision of other methods could be an interesting avenue for further study, especially when considering instruments with higher sampling resolution, but is outside of scope here.

L444-445: Please combine them into one paragraph. -- done

L454: Why does the real uncertainty can only increase? As the effective wind speed aims to minimize the difference between estimates and true emission rates, the turbulence effect can be included there.

As mentioned in the reply to general comment 3, the effect of turbulence is random, and can typically not be reproduced in simulations. Therefore, even if a dedicated simulation is run for any given real-world case, the uncertainty estimated like in this study is expected to be conservative. Additionally, in real-world applications, additional sources of uncertainty have to be also included (e.g. background estimation, instrument noise and/or bias).

**Response to Review #2**

The study of Galkowski et al. addresses the challenges of quantifying emissions from single plume images from airborne or space-borne remote sensing instruments in the presence of turbulence. The problem arises from the stochastic nature of turbulence, which adds uncertainty to flux estimates that is very difficult to mitigate. Even the most realistic and highest-resolution model can only reproduce the statistics of turbulence, but not the exact state of turbulence at any given time as observed during a satellite or aircraft overpass.

Several previous studies have taken note of this issue but have not addressed it systematically. The study presented here is therefore a very valuable and timely contribution to the topic.

We thank the reviewer for the recognition of the importance of the issue.

However, the study has a few issues described below under "major points" that need to be addressed before it can be considered for publication.

**Major points**

• The introduction is too long and not well tailored to the study. It explains at length the process of national inventories and the usefulness of top-down methods, but it falls short in explaining the fundamental issue associated with the turbulent nature of plumes and what this study exactly contributes to the problem. I see little value of sentences like those on lines 21-24 or 37-45 for the scope of this study.

We have reworked the introduction based on the reviewer's suggestion. We have deleted the sentence in L21-24 and merged the first paragraph with the second. We have deleted the paragraph in L37-45 entirely.

There is also quite a few unnecessary filler words like "moreover" or "nevertheless" and awkward formulations like "multiple adverse effects across numerous domains" or "the availability of accurate data is insufficient to provide estimates with low enough uncertainty". Low enough for what? What are "robust studies" (line 54)? Do space-borne platforms really provide "accurate information" (line 50)?

We went through the text again and tried to make it more concise. We focused on the reviewers remarks and modified descriptions mentioned explicitly. We hope that the revised text is easier to follow.

• Section 2.3 introduces the tagged tracers, but it is not explained at this point why they are used and what additional information can be obtained from them. Only stating that they are used "to study the effects of atmospheric turbulence on source estimation inference" is not enough. The purpose of these tracers becomes clear only later, but it should already be motivated here or in the introduction.

We have added a paragraph to section 2.3 that explains the purpose of tagged-tracer usage. It reads:

Using these short puffs allows us to evaluate the impact of large eddies interacting with the tracer at the point of emission as well as during their advection to further downstream areas. For this we specifically calculate plume centroids to follow the motion of each puff (details in Section 2.8.1), as well as the wind speed during the time of the tracer release, which directly impacts the initial dilution of the tracer dilution when emitted into the atmosphere (detailed in section 2.8.2).

• The utility of wind speed at emission point and time introduced in Section 2.8.2 is not clear and not sufficiently well motivated. The apparent emission at location x\_i is determined by the total mass of the tracer at this cross-section and the effective wind speed (Eq. 3). Why should this quantity be related to the wind speed at the emission point? This is not explained at all.

We have added the following description at the beginning of the 2.8.2 section:

We propose that there is a connection between the local wind at the emission point and time and the apparent emissions estimated downwind. That connection is established at the moment of emission and remains in the advected CO2 signal over distances larger than the eddy scale. Air parcels under low local wind would be loaded with higher mole fractions, as the dilution into the atmosphere is lower, while under higher wind speeds the dilution into the atmospheric air parcel is larger. We further argue that this variability of mole fractions persists in the downwind advected plumes, causing variability in the apparent emissions reported in the measurements.

Deviations of the wind at the emission point from u\_eff could simply be used to describe the magnitude of turbulent fluctuations. Approximately the same turbulence should be present at all downstream locations  $x_i$  where the apparent emission is determined. This fluctuation could thus be used as a measure for the variations in the estimates of apparent emissions. However, with increasing distance downstream, the plume becomes wider and thus the flux is no longer determined by the wind at a single location but by the winds along the whole cross-section (which may extend over multiple turbulent eddies).

We agree that the described deviation from  $u_{\rm eff}$  could be used to estimate the magnitude of turbulent fluctuations. However, the method proposed by the reviewer assumes that the variations in the apparent emissions are only linked with the turbulence at the location of the measurement (i.e. taking the cross-section in the CSF method), whereas we postulate that a significant portion of variability is determined already at the emission point and persists over longer distances.

Our experiment additionally demonstrates how the puff structures are organized, rather than chaotic, with the centroids maintaining their positions relative to each other throughout their journey in the advected fields.

What could potentially be interesting is to link the wind speed at the emission point to the mean transport speed of the puffs from their release to the locations  $x_i$ . Initially, they are identical but with increasing distance the correlation will likely be lost.

Figure 2. Mean wind speed of the puff tracer centroid calculated from the time of release to the moment at which each puff crossing the Y-Z plane at a selected X value (coloured lines).

Thank you for the suggestion. Indeed, it was an interesting aspect to consider. Figure 2 (this document) shows the mean wind speed from the time of release to the point at which each puff (numbered 1-60, on the x-axis) reached a certain distance from the emission point. The different coloured lines indicate the distance travelled, from 5 km to 35 km in 5-km increments. The general upward trend from left to right can be explained by the slight increase in mean wind speeds over the course of the day, as puffs with lower numbers were released earlier.

The correlation of the initial wind speed with the mean wind speed across increasing distances does appear to become weaker, but is not lost entirely on these spatial scales. The variability in the mean wind speed does, however, decrease as the averaging distance increases. Generally, the order of the lines remains the same for locally low wind speed anomalies (with the 5-km red line being the lowest and the 35-km magenta line being the highest, e.g. puffs 10-16) but is reversed for locally high wind speed anomalies (most notably around puffs 37-40). There are fewer high anomalies because of the gradual wind speed increase over the course of the day.

• An aspect that would have been useful to explore is whether the impact of turbulence diminishes with increasing distance from the source. This is to be expected because the dispersion of the plume reduces spatial gradients (which reduces turbulent cross-gradient fluxes) and because the lateral extent of the plume increases with distance, such that the plume extends over multiple eddies in across-plume direction. As a consequence, it might be

beneficial to estimate emissions only from cross-sections at larger distances. On the other hand, a lower number of cross-sections reduces the advantage of averaging.

We partially addressed this issue in the study by comparing the estimates collected from cross-sections evaluated in X ranges 2-22 km vs 2-40 km. The results are given in L337-L340 and are summarized in Table 1. The discussion of these results in L375-L381 focused on the increase in uncertainty and not the diminishing impact of turbulence with distance.

We have added the analysis of fluxes calculated using cross-sections over 20-40 km. We have added an extra entry in Table 1, and a description text in the Results section, after L340. In the discussion, we have added the following text after L381 (RL430).

The reduction in the uncertainty when the longest plume segment is analysed is caused by an increase in the overall number of observations. However, it is likely also related to the gradual dissipation of the correlated structures in emitted  $CO_2$ . Reduced variability in  $\Phi(x_i)$  can be observed at distances larger than 20 km (Fig 5). To investigate this further, we have calculated the mean apparent emission using cross sections from 20-40 km, applying the same method. This yields a mean emission rate of 38.4 Mt yr-1, with an uncorrelated uncertainty of 0.6 Mt yr-1, less than half the size as when cross-sections are sampled between 2-22 km. When correlations are included, the uncertainty estimate is also lower, yielding 3.3 Mt yr-1 (8.7 % relative). This is achieved despite the increased  $d_{indep}$  (4.6 km vs. 3.6 km for 2-22 km). Our interpretation of this is that our method is still able to recognize the persistent structures in the downwind plume even though the variability of individual puff contributions becomes smoothed out with distance.

The reduction in mean emission uncertainty between estimates for the near (2-22 km) and far plume segments (20-40 km) suggests that it is beneficial to apply CSF further downwind from the source, where the initial field variability is partially reduced. However, in real-world applications, the effective measurable signal may go below the detection limit, especially for weaker sources. Analysing at an increased distance might, in addition, cause the assumption of the uniform effective wind speed to become less realistic due to spatial and temporal variability in the winds. This can be caused by i) synoptic changes over the analysed distance, ii) diurnal-cycle-driven changes in wind patterns and iii) local channeling flows. All of these will cause the error to accumulate with time and thus distance, potentially negating the positive effect of weaker spatial correlations in the observed signal.

**Minor points and corrections**

- Line 3: delete "status of the" (repetition of "status") -- deleted
- Line 7: "Realistic assessment" not "Realistically assessment" corrected according to comment from Reviewer #1. See above.
- Line 8: Suggestion: use ".. the impact of the stochastic nature of .." instead of "stochastic impact" agreed
  - Line 9: "on the estimation" not "estimations". We've reworked the sentence. It now reads:

Here, we examine how the stochastic nature of daytime atmospheric turbulence affects the estimation of CO2 emissions from a lignite coal power plant in Belchatów, Poland. For this

investigation, we use a high-resolution ( $400 \text{ m } \times 400 \text{ m } \times 85$  levels) atmospheric model set up in a realistic configuration.

**Line 11: turbulent structures in a plume are not "persistent"**

Agreed in principle, however please note that we're not talking about "turbulent structures" here, but only "structures":

We show how the persistent structures in the emitted plumes (...)

While small scale turbulent structures are fully stochastic, the structures in the dispersed signal preserve their shape and pattern through many eddies, and are long-lived as compared to the eddy-scale of turbulence. To make this clearer, we've reworked the sentence.

We demonstrate that persistent structures in the downwind concentration fields of emitted plumes can cause significant uncertainties in the retrieved fluxes on the order of 10 % of the total source strength, when the commonly used cross-sectional mass-flux (CSF) method is applied with short distances between individual estimates.

Line 12: "... method, on the order of " (replace . by ,) – agreed see above.

Line 14: These are just temporally tagged tracers. I don't see the novelty. This is by far not the first study using tracers tagged by the time of emission.

We did our best to find studies using a similar technique applied to atmospheric tracer modeling. We scanned databases for "tagged tracers", "tracers tagged by time of emission", "temporally tagged tracers", "pulse tracer emissions", etc., however we were unable to find similar papers addressing this problem. We agree, however, that there is significant similarity to applications of Lagrangian models, which must run emissions for distinct releases (essentially realizing the same concept) to predict downwind mole fractions. A study by Stohl et al. from 2011, where a Lagrangian model was used in the forward mode to simulate ash concentrations based on emissions tagged by altitude and time was the closest to our application. However, in the analyses these forward simulations are used in a Bayesian inversion similar to earlier works by Gerbig et al. (2003) and Lin et al. (2004) – both cited in our manuscript.

We have not seen the method applied in analysis like ours in 3D Eulerian simulations of atmospheric pollutants. Nevertheless, we propose to remove "novel" from the statement in the abstract, so that it now reads:

Furthermore, we applied temporally-tagged tracers for the decomposition of the plume variability into its constituent parts. These tracers helped us to explain why spatial scales of variability in plume intensity are far larger than the size of turbulent eddies – a finding that challenges previous assumptions.

Line 15: ".. and TO explain why" – Revised, see immediately above.

Line 20: "represented by A 1.5°C temperature increase" – corrected.

Line 28: Is "accurate" not enough instead of "accurate, precise ..."? – agreed

- Line 45: Unclear what you mean by "have also been recognized" sentence removed.
- Line 47: Delete "accurately", not necessary. removed
- Line 50: Delete "accurate". It is clear that they aim to provide accurate information, but they do not necessarily achieve that, and it remains unclear what "accurate" means without a definition. deleted
- Line 58: Another important reference for urban mass balance approaches is Cambaliza et al. (2013, <a href="https://doi.org/10.5194/acp-14-9029-2014">https://doi.org/10.5194/acp-14-9029-2014</a>).

Thank you for pointing out this study, reference added.

Line 61: Not very clear what is meant by "some instrumentation", and "applied" is not the right verb. There are important airborne remote sensing instruments used for emission quantification missing in the references, notably AVIRIS-NG and MethaneAir. Furthermore, it's not clear that by "orbiting platforms" you mean satellites.

We have added references to both AVIRIS-NG (Thorpe et al. 2016, <a href="https://doi.org/10.1016/j.rse.2016.03.032">https://doi.org/10.1016/j.rse.2016.03.032</a>) and for MethaneAir (Chulakadabba et al., 2023, <a href="https://doi.org/10.5194/amt-16-5771-2023">https://doi.org/10.5194/amt-16-5771-2023</a>)

The sentence containing 'orbiting platforms' has been revised, it now reads:

Although remote sensing instruments installed on airborne platforms have been used successfully for this purpose in the past (Krings et al., 2013; Thorpe et al., 2016; Krautwurst et al., 2021; Wolff et al., 2021), satellite observations offer a distinct advantage due to their global coverage and lower cost per observation.

- Line 63: "In fact" unnecessary. -- deleted
- Line 64: Change to "OCO-2/3 observations were used" added
- Line 68: CarbonSat was planned as an Earth explorer mission, not an operational mission.

  thank you for the clarification. This is included in the response to the next comment.

Line 70: Maenhout et al. (2020, https://journals.ametsoc.org/view/journals/bams/101/8/bamsD190017.xml) would be the better reference

Reference added. The fragment now reads:

Early steps towards such a system were taken through the proposed Earth Explorer mission CarbonSat (Bovensmann et al., 2010). This work was subsequently expanded and resulted in the design and approval of CO2M (Copernicus Anthropogenic CO2 Monitoring Mission), a constellation of satellites that are to be launched within the current decade (Sierk et al., 2021), and will form the backbone of the operational system with the CO2 emission Monitoring and Verification Support (MVS) capacity, as described by Janssens-Maenhout et al. (2020).

Line 75: "applications IN the past". And change to "more recent developments"

Agreed, text altered taking the comment into account. Please also see above in the response to the related comment by Reviewer 1.

Line 81: Why does variability stem from the "estimation method"? Please be more specific.

This comment was addressed together with a remark from Reviewer 1 that concerned the same line. See above.

Lines 85-86: Gerbig et al. and Lin et al. are good references for some problems of transport uncertainty (PBL height, uncertainties in wind speed and direction (not turbulence!), but they are not good references for addressing issues related to the stochastic nature of turbulence.

We respectfully disagree. Turbulent eddies are specifically mentioned and their impact is estimated in the analysis of the spatial variability of atmospheric CO2 in Gerbig et al., 2003a and Lin et al., 2004. Please note that these papers are not good references for transport uncertainties such as PBL height or wind speed and direction.

Gerbig, C., J. C. Lin, S. C. Wofsy, B. C. Daube, A. E. Andrews, B. B. Stephens, P. S. Bakwin, and C. A. Grainger (2003a), Toward constraining regional-scale fluxes of CO2 with atmospheric observations over a continent: 1. Observed spatial variability from airborne platforms, J. Geophys. Res., 108(D24), 4756, doi:10.1029/2002JD003018.

Please also note that in the original manuscript the citation to Gerbig et al. (2003) incorrectly cited a companion paper. This has been corrected.

Line 87: Similar to what? Similar to Gerbig et al. or similar to the present study?

Similar to the Gerbig et al, 2003a and the Lin et al. 2004 studies, where the spatial variability was analysed using variogram analysis (which was picked up by Fuentes et al., 2024). The text has been changed to:

A similar approach to the one used in those studies (...)

Line 92: Change to "insight into the mechanisms". What is a "detailed mechanism"? Also change to "to shed light". – changed as suggested

Line 93: "are using ... simulations" (not simulation) – corrected, merged with remark from Reviewer 1.

Line 97: Again "detailed" is unnecessary. – deleted

Line 100: "on the planet" is enough. – agreed, merged with a comment from Reviewer 1

Line 119: change to "at scales ranging from global to local" – agreed, see comment from Reviewer 1.

Line 121: "Employed" rather than "deployed" the WRF model. – changed as suggested:

Line 150: It would make things clearer if you write "time-varying Cartesian coordinate system" – thank you for the suggestion, added

Line 159: "single point source" rather than "single-point source". – changed

How many sources are there in reality, and why is it justified to treat them as a single source at 400 m resolution?

At the plant there are 3 sources, each about 400 m apart: two primary high stacks (300 m high) through which CO2 is discharged from power generation units B2-B12 (unit 1 was decommissioned earlier) and a coolant tower that is connected to a modern B14 unit, in which flue gas from the flue gas desulphurization (FGD) system is discharged.

Nassar et al., 2022 analyzed the data from the European Network or Transmission System Operators for Electricity (ENTSO-E) Transparency Platform and found that block B14 was not operational on 4 April, 2020, the date of our analysis. That means that only the tall stacks were operational that day. These two are exactly 330 meters apart, therefore we treat them as a single source. We have added the following to the manuscript:

Three individual stacks were responsible for  $CO_2$  emissions at BPP in 2020. Nassar et al. (2022, Table 4) used publicly available data and found that only blocks B2-B12 of the power plant we operational on 10 April 2020, our date of interest. These blocks emit  $CO_2$  through two tall (300 m high) stacks located 330 m apart. In our model we combined both into a single point source, as our horizontal grid size is 400 m

Line 165: Change "are and" to "are" – done

Section 2.4: Why is the period 9 - 10 April 2020 used in this study? This should be better motivated.

The experiment was inspired by informal discussion that took place at an early stage of the Fuentes-Andrade et al.(2024) study, when data from April 2020 were presented. We used this date as the result clearly presented the effect we wanted to investigate.

We have added the following sentence:

This date was selected as a good candidate as OCO-3 observations from that day displayed characteristic variability of apparent emissions that we investigate in this study.

Line 178: On line 148 it was stated that output is saved at 5 minute resolution, but here it is 1 minute. – Thank you for spotting this. One minute is correct. Five-minute output was used during early runs. This text fragment was written then and should have been updated.

We removed the sentences around line 148 to avoid repeating information.

Section 2.5: Column average dry mole fractions are frequently used because they are NOT proportional to the total mass of the tracer. Variations in surface pressure and topography are largely eliminated when using dry air mole fractions, whereas they affect the total mass of the tracer.

Thank you for pointing this out. We have corrected the text:

The column-averaged dry-air mole fraction, commonly used in remote sensing measurements, is a scalar quantity that integrates trace gas abundances across the whole atmospheric column. It offers advantages over reporting in mass units, as it reduces the influence of surface pressure and topography on the retrieved signals

Line 190: To make clear that n\_d is not a density (moles m-3), it should be explained that n\_d is the number of moles of dry air in the grid cell at x\_i, y\_j, z\_k. Otherwise Equation 4, which divides by the grid cell area A would be wrong. – agreed, the line now reads:

" $[n_d]_{ijk}$  is the number of moles of dry air in the grid cell at  $x_i$ ,  $y_j$ ,  $z_k$ "

Line 192: change to "with THE x-axis .." – added

Line 194: replace "all the way to orbit" by "to the top of the atmosphere". – replaced

Line 197: I don't agree that it is not necessary to extend beyond the model top at 50 hPa, because the total number N\_d of dry moles in the vertical column is about 5% larger when extending beyond 50 hPa, which has the effect that the mole fraction enhancements calculated following Eq. 1 become 5% lower (because the weights are proportional to 1/N\_d, see Eq. 2).

Point taken. This is relevant for the calculation of  $XCO_2$ , which is only used for visualization purposes in Figures 3a and 4. We have revised these figures appropriately. It does not affect any of the calculations using  $\Delta\Omega$ , as these are defined in mass per unit area, which is not affected by the height of the column, as long as it contains the whole plume. We removed the text from L194-197, and inserted the following sentence in L191 to clarify:

Because our model top was set at 50 hPa, we applied a correction to account for the missing atmospheric mass when calculating the weights.

Equation (3): This equation is only correct if turbulent flux along the x-axis is negligible compared to the advective flux represented by u\_eff. Section 3.4 in Conley et al. (https://doi.org/10.5194/amt-10-3345-2017) provides an excellent description of turbulent flux terms versus advection. In case of low wind-speeds, the contribution of turbulent fluxes may not be negligible. Furthermore, it should be more clearly explained that Eq. 3 describes the flux through a plane perpendicular to the wind direction, and therefore integrates along the y-axis.

We agree, and thank the reviewer for this reference. In our study, we have followed the selection criteria set in Varon et al. (2018), which stated that the lower limit of wind speed that fulfills this condition is 2 m/s. Our analysis shows that the average ueff was close to 3 m/s. Nassar et al. (2022) and Fuentes Andrade et al. (2024) both assumed that for 4 April 2020 the conditions are fulfilled.

We have expanded and modified the text after L218 (RL223) as follows:

It should be noted that the equation is true when turbulent flux along the X axis is small compared to the advective flux characterized by  $u_{\rm eff}$ . An excellent overview of the turbulent and advective flux terms is available in Conley et al. (2017), who show that when winds are close to and below this threshold, upwind-directed fluxes may cause overestimation of the scalar source strength for near-surface point sources. Varon et al. (2018) have argued that for a typical turbulent day this condition is met when wind speeds are  $2 \text{ m s}^{-1}$  or higher and used

this value as a lower limit of the applicability of the CSF method. We follow the approach of previously published measurement-driven studies that included the analyzed case (Nassar et al., 2022; Fuentes Andrade et al., 2024) and assume that the turbulent flux component can be neglected in the downwind areas.

Regarding the second part of the comment, we have expanded the description leading to Eq. 3. (L199-201) in the following way:

By assuming that the mass of the tracer is conserved (true in the case of long-lived greenhouse gases advected over short distances), emission rates at the source can be inferred by integrating the tracer mass elements passing through a plane perpendicular (i.e. along the Y axis) to the wind direction, at a certain distance x downstream from the source. This can be described mathematically as: (...)

Line 205: I would call DeltaOmega(x,y) the column-integrated enhancement of CO2 rather than column-averaged. Note that this quantity is not "integrated along the y-axis" (only Phi(x) is).

Accepted. Clarified the sentence by removing "integrated along the Y axis"

Line 206: "mimic" is probably the more appropriate term than "reproduce" – agreed

Line 208: What formula did Varon et al. use for u\_eff? Actually, the way u\_eff is calculated is described in more detail on lines 227-230, and I doubt that Varon et al. did it exactly in the same way.

Indeed, we have used a very different definition. In Varon et al. (2018), the authors used an idealized WRF-LES modelling framework to infer  $u_{eff}$  as a function of the  $U_{10}$  at the source location. The relationship they obtained is given in formula (12) of their work:

$$u_{eff} = \beta * U_{10}$$
,

where  $U_{10}$  is the 10m wind speed perpendicular to the plume from the LES model they used, sampled at the source location, and  $\beta$  is a scaling factor depending on the measurement instrument precision ranging from 1.3 to 1.5. They determined this using their modelling framework, using known source rates injected in an ensemble of simulations covering a wide range of wind speeds and directions over an area containing a single point source.

To apply Varon's approach relies on precise wind speeds at 10 m being available. Such measurements are not available for the BPP, and we have decided not to use the modelled value in order to follow procedures as applied by the measurement community, and used a more direct approach in line with the more recent studies of Kuhlmann et al. (2021) and Nassar et al. (2022), where ueff is estimated by sampling global datasets at altitudes close to the plume vertical level, as described in the paragraph at L220—230.

Regarding the second sentence in the comment: In L220, it was not our intention to imply that our approach was similar to that of Varon et al. (2018), but rather to emphasise that we use a single ueff value in formula (3) rather than using wind speeds estimated at different distances from the source.

We have clarified the text by removing the information about the ueff calculation near L208, part of which repeated information from the paragraph at L220-230, where we have clarified the description:

We calculate the effective wind speed  $u_{eff}$  from wind fields sampled at altitudes close to the emission altitude. The appropriate vertical level is selected from our input emission profile. This approach is a hybrid of those used in the recent studies of Kuhlmann et al. (2021) and Nassar et al. (2022). In the first study, the mean wind speed was calculated from the model output winds, weighted by relative emission strength. (...) Here, we calculate  $u_{eff}$  as an average of wind speed values at altitudes between  $H_{eff} \pm 2\sigma_H$  (200 m–600 m). As we aim to mimic processing as performed in studies using actual satellite imagery, we assume a constant  $u_{eff}$  throughout the area of interest despite having access to complete modelled wind fields. We also spatially average the wind speeds over a square area of  $\pm$  20 km around the emission point, which mimics the effect of using a coarse-resolution reanalysis wind dataset like ERA5 (as in Nassar et al.) that does not represent variabilities on smaller scales

Equation 4: This equation results in units of g/m2, not kg/m2. – Molar mass of CO2 now given in kg mol-1.

Line 214: change to "Applying the above to Eq. 3". Change x to  $x_i$  i – changed

Line 216: No need to state again that "other symbols are as before". – removed

Line 226: Not clear to me what is meant by "Gaussian plume assumptions". This assumption doesn't seem to be necessary to simply compute a plume centreline.

We meant that Nassar used a Gaussian plume model, which assumes that the centerline remains at the height of the stack unless buoyancy is taken into account. Nassar et al. (2022) used a plume rise of 250 m following Brunner et al (2019).

We have clarified this fragment in the following way:

In the second study, Nassar et al. used a Gaussian plume model to simulate the plumes from BPP, with the plume centreline set at 250 m above the stack height to represent the additional plume rise ( $H_{\rm eff}$ ), following Brunner et al. (2019). Subsequently, they used winds from reanalysis datasets extracted at the same height over the emission point to calculate  $u_{\rm eff}$ .

Line 228: Change to "affecting u eff" – this line was altered, see comment to L208.

Line 231: It is unclear at this point what quantities will be correlated against what other quantities and thus why the definition in Eq. 6 is useful. Why do you "also" make use of the apparent emission anomaly? What other things are you using?

Changed - "also" is dropped. We have altered the text according to reviewer's suggestion to clarify where the correlation analysis is used, referring to section 3.2.

Line 238: Change to "dependent on the effective number" – done

Line 249: Change to "along THE x-axis" – done

Line 251: This sentence is not quite correct and should be deleted. A particle in a Lagrangian particle dispersion model is not similar to a plume centroid. Particles rather span the whole plume.

We agree that the sentence is not precise enough. What we meant is that the individual puffs themselves can be seen as analogous to the particles (or air parcels) in Lagrangian models. On a larger scale, the plume centroids as defined here would be very similar to particle locations.

We deleted the sentence as per the reviewer's suggestion.

Equation 9: Why do you use lower case "c" rather than upper case "C" as in Eq. 4.? Is "c" a different quantity? Is it concentration rather than dry mole fraction. To describe a real center of mass, c\_i,j,k in Eq. 9 should actually be the mass of the tracer in grid cell i,j,k rather than the concentration or dry mole fraction.

Uppercase C should be used, thank you for pointing it out. Indeed, we have defined the plume centroid without using mole fractions rather than mass. This is specifically stated in L 252-254:

"We define a plume centroid as the first moment of the distribution of the tracer's mole fraction, thus approximating each tracer's centre of mass".

We decided not to use the centre of mass to maintain mole fraction as the primary quantity. The difference between mass- and mole-fraction-based centroids was tested and shown to be negligible in our simulation.

Lines 257-261: What do you mean by "density function of puff-centroids"? I did not understand these sentences at all. I could only guess what they mean after having read the rest of the paper.

We apologise for the lack of clarity. We have expanded the description in the following way:

To investigate the relationship between the number of plume centroids at a given distance  $x_i$  and apparent emissions  $\Phi(x_i)$ , which is related to the meandering of the plume, we use the puff centroid density  $r(x_i)$ , calculated for each  $x_i$  as the sum of plume centroids falling within x values in the range  $[1/2 (x_{i-1}+x_i), 1/2 (x_i+x_{i+1})]$ . Due to the low number of centroids imposed by the computational constraints, we cannot estimate r directly at the full resolution of our interpolated grid. Instead, we follow a two-step procedure: first we bin the centroids at a reduced resolution of 2 km, and then we use a cubic spline interpolation to obtain the centroid density at a full 200 m resolution.  $r(x_i)$ , and its spatial average  $\bar{r}$ , are then used to calculate the normalized anomaly of the centroid density,  $\lambda_c$ , analogously to Eq. (6).

Line 279: "of each" puff? – added

Line 282: Why Xc and not XC as in Equation 1? – thank you, corrected

Line 288: Why "also" and not simply "we correlate"? – deleted

Line 294: The simulations alone cannot demonstrate that the day was typical, because they extended over 2 days only.

Here, we wished to underline that the simulations realistically reproduce conditions characteristic of a mid-latitude spring day with clear sky. We have altered the sentence in question:

"The simulated meteorological conditions show a nocturnal stable atmosphere evolving into a turbulent PBL over the course of the morning (...)"

Line 300: Delete "orbiting" – done

Figure 3: The vertical dispersion (lower left panel) would be better visible when integrating the tracer in across-plume direction rather than showing it only along the centerline.

Agreed. The plot was redone as suggested, Figure caption and text were altered accordingly.

Line 332: Change to "due to THE moderate sample size" – added

Line 360: What characteristics should one expect? – We meant the ones listed in the next sentences. "Expected" has been deleted to avoid confusion.

Lines 360 - 361: The sentence "The estimated cross-section emissions show typical features" makes little sense to me.

We politely disagree. We believe that a pollutant plume is recognizable not only to scientists involved in their studies but also to the general public. By this sentence (and the "expected" deleted above) we wish to convey that the plume is typical, without displaying any unexpected features.

Line 367: Change to "is remarkably similar to that" – changed as suggested

Line 371: The result that independent estimates are obtained only every 3.6 km is very relevant in the context of satellite observations. Satellites with coarser resolution will lose useful information that could have been exploited at higher resolution. How do the 3.6 km relate to actual satellite missions?

We thank the reviewer for this question. Indeed, this is highly relevant, and may be a limiting factor in the retrieval of emissions from satellite missions with imaging spectrometers with a coarser ground pixel resolution, such as TROPOMI, with a resolution of 5.5 km x 7 km, or the upcoming Sentinel-5, with a resolution of 7 km x 7 km. In this case, useful information about the plume structure is being lost. In contrast, many other existing (GHGSat, MethaneSAT) and upcoming (CO2M) missions are operating at higher spatial resolutions (on the order of 20 m, 200 m, and 2 km, respectively). When analyzing turbulent plumes measured by sensors at higher spatial resolution, the autocorrelation length needs to be taken into account when estimating the information content of the measurement, and when calculating the uncertainty of the emission estimate.

Line 374: The realism of the WRF-Chem simulations was also evaluated in the Brunner et al. study. What was the conclusion in that study?

The Brunner et al. (2023) study found that representing the turbulent structures of power plant plumes required model resolution of 1 km or better, and that model resolution had a larger impact than differences in the treatment of turbulence e.g. between NWP models and LES models. A WRF-GHG simulation at 400-m resolution, employing a very similar setup to the one in this study (only notable difference is the PBL parameterization), was compared against remote-sensing and in-situ trace gas measurements in the vicinity of BPP on 7 June, 2018, and near the Jänschwalde power plant on 23 May, 2018. The WRF-GHG showed "remarkable consistency" with the measurements of the turbulent plume at BPP, but underestimated the vertical plume extent at Jänschwalde. One of the differences was the time of day of the measurements: at Jänschwalde the data were collected in the morning, before turbulence had had time to fully develop, whereas the measurements at BPP were made in the early afternoon. While this is not a guarantee that the simulations of the BPP plume on 10 April 2020 are equally accurate, the model has been shown to be capable of realistically representing the turbulent plume from BPP.

In L374 we have added:

(...), confirming the high capability of WRF-GHG, previously reported by Brunner et al. (2023).

Line 377: Change to "section of THE plume" – changed

Line 379: The factor four has little importance, because it entirely depends on the number of cross-sections taken, which in the present study is determined by the model resolution. Here this factor is presented like a general uncertainty amplification factor.

We thank the reviewer for this insightful comment.

Indeed, the factor four is not general and will not appear if the cross-section configuration is altered. To be precise, it is not so much dependent on the number of cross sections, but rather the distance between the cross sections. As long as these are taken at distances corresponding to the length scale of independent observations, a high number of cross sections could theoretically result in lower uncertainty. In practice, this is impossible, as after e.g. 200 km the plume from a power plant, even one as large like BPP is likely to have dissipated to the point that it is no longer measurable.

We have revised and expanded discussed fragment in the following way:

When the correlation of observations is taken into account, the uncertainties of the emission estimate become significantly higher, in our case increasing by a factor of four. The extra uncertainty stems from correlation in the  $\Phi(x_i)$  that occurs due to turbulent dispersion, and it reduces the number of effective observations when cross sections of CSF are selected at distances lower than  $d_{indep}$ . This minimum distance is imposed by the physical properties of the system, and uncertainty from a single scene cannot be reduced with an increasing density of cross sections. A larger number of truly independent samples could theoretically reduce the uncertainty, but for a single scene this may mean sampling at distances where the signal-to-noise ratio becomes too low, or where other assumptions of the CSF method (especially with regards to wind) are no longer fulfilled.

Line 392: Unclear what is meant by "same subset". Same as what? – The full set of scenes from Fuentes Andrade et al. was meant. Changed to "same set of scenes".

Line 400: A systematic comparison between CSF, GPI and IME methods was recently presented by Santaren et al. (2025, https://doi.org/10.5194/amt-18-211-2025).

We thank the reviewer for pointing out this interesting study. We have added the reference and the discussion to of findings by Santaren et al. (2025). The modified text is provided under the response to the comment concerning the same fragment, made by Reviewer 1 (see above).

Line 404: What about ".. as generally fewer observations over shorter distances are available"? – agreed, thank you for the suggestion

Line 413: Not quite clear whether vertical or horizontal advection is meant here by "atmospheric advection".

Changed to:

The first is the uneven vertical distribution caused by differences in horizontal advection at different altitudes.

Line 454: Use "can only be higher" rather than "increased". Another source of uncertainty is errors in the observations. – changed, and added. The sentence now reads:

When applied to actual observational data, this uncertainty can only be higher, primarily due to imperfect knowledge of the wind fields, inaccuracies in the background estimation, as well as errors in the observations.

Line 459: Turbulent structures are (usually) not persistent.

Changed to (RL531):

*It should be noted that the persistent spatial anomaly structure induced by turbulence (...)*

---

## Author Response (AR2)

**Response to technical corrections**

What follows is the response to the request for technical corrections from Reviewer #2, available via MS system of ACP journal (and later, after acceptance and submission, to the public). The paper that this concerns is:

"Impact of atmospheric turbulence on the accuracy of point source emission estimates using satellite imagery", by Galkowski et al., available at:

https://egusphere.copernicus.org/preprints/2024/egusphere-2024-2792/

The original text from the reviewer is given in black, while the responses from authors are in blue. Text cited from the manuscript is in *italics*. In this document, we refer to the original manuscript using Lx notation (consistent with the reviewer's comments), while an R index is added when the revised manuscript is cited. For example, L15 refers to the original manuscript, while RL15 refers to the revised version.

**Note from authors:**

Before addressing specific comments, we would like to offer our sincere thanks to both reviewers for taking time in review and in reading our response. All comments are highly appreciated as they have made our study better in many respects. We have added a comment in the "Acknowledgments" section stating so.

**Remarks from referee #1**

Note: author made no comment and "accepted as is".

**Remarks from referee #2**

The revised manuscript is greatly improved:

- The introduction was too general and is now much better tailored to the scope of the study.
- The role of the tagged tracers is now better motivated
- The basic hypothesis that the initial wind conditions at the point of release critically determine the turbulent structure of the plume and correspondingly the emission estimates at different downwind cross-sections, is now better substantiated. This seems to me to be the most important conclusion of the study.

My remaining comments have been addressed appropriately.

A few remaining corrections:

- Line 167: "the power plant we operational" -> "the power plant were operational" -
- Line 190: the second "dilution" should be deleted deleted
- Line 429: "especially with regards to wind" should be explained better. Which assumption regarding winds is not fulfilled? The assumption of stationarity?

The sentence in question (RL427) has been expanded, it now reads:

A larger number of truly independent samples could theoretically reduce the uncertainty, but for a single scene this may mean sampling at distances where the signal-to-noise ratio becomes too low. Another risk at large distances is that the assumptions of the CSF method, specifically regarding the uniformity of the wind speed and direction, may no longer be fulfilled.

- Line 524: In table 2 it was 14.2%, not 14.3% - thank you for spotting this, it is corrected

**Other corrections**

Apart from the technical corrections requested by Referee #2, we took the opportunity to fix minor typing errors that evaded detection. As these are very minor, we do not list them here. These can be found in the tracked version of the manuscript. We have also altered the Acknowledgment section, adding thanks to the reviewers, and adjusting minor language issues.